# Language Models Can Improve Event Prediction by Few-Shot Abductive Reasoning

**Xiaoming Shi**[1]    **Siqiao Xue**[1]    **Kangrui Wang**[3]    **Fan Zhou**[1]    **James Y. Zhang**[1]
**Jun Zhou**[1]    **Chenhao Tan**[2]    **Hongyuan Mei**[3]
[1]Ant Group    [2]UChicago    [3]TTIC
`{peter.sxm,siqiao.xsq,hanlian.zf,james.z,jun.zhoujun}@antgroup.com`
`chenhao@uchicago.edu`   `{kangrui,hongyuan}@ttic.edu`

## Abstract

Large language models have shown astonishing performance on a wide range of reasoning tasks. In this paper, we investigate whether they could reason about real-world events and help improve the prediction performance of event sequence models. We design LAMP, a framework that integrates a large language model in event prediction. Particularly, the language model performs abductive reasoning to assist an event sequence model: the event model proposes predictions on future events given the past; instructed by a few expert-annotated demonstrations, the language model learns to suggest possible causes for each proposal; a search module finds out the previous events that match the causes; a scoring function learns to examine whether the retrieved events could actually cause the proposal. Through extensive experiments on several challenging real-world datasets, we demonstrate that our framework—thanks to the reasoning capabilities of large language models—could significantly outperform the state-of-the-art event sequence models.

## 1 Introduction

Prompting large language models (LLMs) such as GPT-3.5 has recently become a standard approach to perform text-based reasoning tasks. In this paper, we investigate their capabilities in reasoning about real-world events and improving event prediction. Particularly, we focus on the problem of modeling sequences of time-stamped events and predicting future events given the past. For example, in the healthcare domain, we would like to model patients' sequences of time-stamped hospital visits and predict their future symptoms given their past diagnosis and treatments. It has been a long-standing and important problem in machine learning. Large language models are potentially useful for advancing solutions to this problem because event sequences are often accompanied with rich text information which large language models excel at handling. For example,

- *Healthcare.* Each hospital visit will have a doctor note summarizing this visit, including the department that the patient visits, the clinical measurements and treatments, and any future medical plans. By reading such textual information, a large language model may be elicited to recall the medical knowledge that it has read during pretraining and then reason about the future hospital visits such as what symptoms or treatments that the patient may have.

- *Political.* Each political event may generate a series of news articles describing the political agents involved in it and discussing its possible influences. A language model reading these articles may recall its knowledge—which is acquired from pretraining—about these agents, their relations, and fundamental principles in politics such that it could reason about future political events.

- Similar scenarios arise in *commercial*, *dialogue*, *finance*, etc.

In this paper, we propose LAMP, a framework that integrates a large language model in event prediction. The overview of our framework is illustrated in Figure 1. Given a history of previous

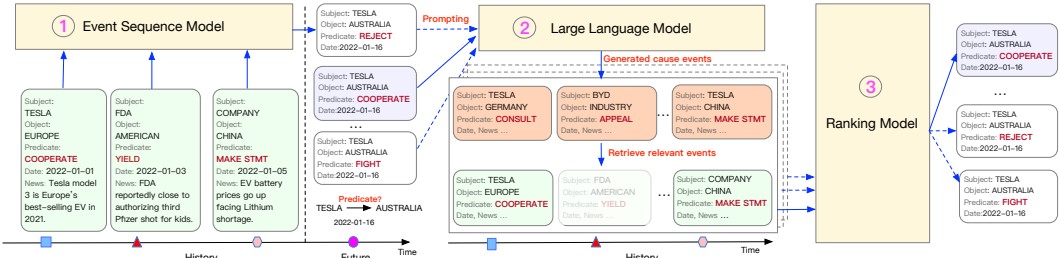

Figure 1: An overview of our framework that leverages a large language model to reason about events. Firstly, an event sequence model proposes predictions: in this example, we predict the predicate of the structured event type given its time, subject, and object. Secondly, a language model suggests cause events, which will pattern-match against actual previous events and retrieve the most relevant. In the end, a neural model learns to assign high scores to the proposed predictions that are strongly supported by the retrieved evidence.

events, we use a pretrained event sequence model to propose predictions on the future events, which are then examined with the assistance of an LLM. The LLM learns to perform abductive reasoning: it is instructed by a few expert-annotated demonstrations, and generates possible causes that may explain the possible occurrence of each proposal. Each generated cause serves as a query to search for similar or relevant events that have actually happened. Then another neural model learns to embed these retrievals and examine whether they could really justify the corresponding proposal.

We are the first—to the best of our knowledge—to integrate large language models into event sequence modeling. Our modeling and prediction framework is general: it can incorporate all kinds of event sequence models and large language models. We experimented with a range of model choices and demonstrate that large language models could indeed help improve the prediction performance of event sequence models. On several challenging real-world datasets, our framework significantly outperforms the current state-of-the-art event sequence models.

## 2 Problem Formulation and Technical Background

Now we give a formal introduction to our problem setting and review the background knowledge.

**Event sequence modeling.** The problem is to model event sequences $(t_1, k_1), (t_2, k_2), \ldots$, where $0 < t_1 < t_2 < \ldots$ are times of occurrence and each $k_i \in \mathcal{K}$ is a discrete event type. The goal is to predict the next event for a given history of events $\mathcal{H}_i = (t_1, k_1), \ldots, (t_{i-1}, k_{i-1})$. Precisely, it consists of two subtasks: the first is to predict the time $t_i$ of the next event; the second is to predict the type $k_i$ of the next event with the knowledge of its time $t_i$.

The standard approach is to build a probabilistic model over the sequences. Such models typically define an intensity function $\lambda_k$: the intensity value $\lambda_k(t)$ is the instantaneous rate that an event of type $k$ occurs at time $t$. Given the function $\lambda_k$, one could obtain the minimum Bayes risk (MBR) prediction of the next event given the history. Particularly, the MBR time prediction $\hat{t}_i$ is

$$\hat{t}_i = \int_{t_{i-1}}^{\infty} t\lambda(t) \exp\left(-\int_{t_{i-1}}^{t} \lambda(s)ds\right) dt \text{ where } \lambda(t) = \sum_{k \in \mathcal{K}} \lambda_k(t) \tag{1}$$

and it could be approximated by averaging samples given by the thinning algorithm (Lewis & Shedler, 1979; Liniger, 2009). The MBR type prediction $\hat{k}_i$ given time $t_i$ is

$$\hat{k}_i = \text{argmax}_k \lambda_k(t_i) \tag{2}$$

The intensity function $\lambda_k$ is typically learned by maximizing the log-likelihood of the model. For a time period $(0, T)$ that contains observed events $(t_1, k_1), \ldots, (t_I, k_I)$, the log-likelihood is

$$\sum_{i=1}^{I} \log \lambda_{k_i}(t_i) - \int_{t=0}^{T} \sum_{k \in \mathcal{K}} \lambda_k(t)dt \tag{3}$$

**Rich text information.** In real-world data, each type $k$ may be represented as a text-based identifier: in the example of Figure 1, each $k$ is one of the possible interactions between the political entities (organizations and individuals) in the G20 countries, which can be represented with a structured name

such as Tesla-cooperate-Australia. In addition, each event may have a text mark $\mathbf{m}$ that contains additional information about the event: in Figure 1, each $\mathbf{m}$ is a news headline about the event (e.g., "EV battery prices go up"). For notation simplicity, we will only mention the mark $\mathbf{m}$ of an event when necessary. While reading such text information, a human may recall their relevant domain knowledge (e.g., influence of battery prices on Tesla and Australia) and increase their estimate on the probability that an event of Tesla-cooperate-Australia happens in the near future. An important way that humans learn such knowledge is reading text such as textbooks, research publications, and news articles. But event sequence models can not directly leverage this kind of information.

**Large language models.** Language models learn by reading text. Over the past years, large language models that have read nearly the entire internet have shown astonishing performance on many challenging tasks such as arithmetic reasoning and multi-turn dialogue (Wei et al., 2022b; OpenAI, 2023). So it seems tempting to pair a large language model with an event sequence model to improve its prediction performance: the language model has consumed a tremendous amount of information that the event model may not have seen but should be able to benefit from.

# 3 LAMP: Large Language Model in Event Prediction

Now we present our LAMP framework, in which an LLM is leveraged to enhance the prediction process of an event sequence model. As shown in Figure 1, LAMP has three key components:

- A base event sequence model. This model is pretrained and we use it to propose candidate predictions. Section 3.1 is the discussion of this phase.
- A large language model. Its duty in the framework is to perform *abductive reasoning*, a form of logical inference seeking the most plausible explanations for a given observation (Russell & Norvig, 2010). Particularly, the language model reads each proposed prediction and suggests possible cause events for it. Then we pattern-match each LLM-generated cause against the actual previous events in the history, and retrieve those which are most similar. Section 3.2 discusses this phase.
- A ranking model. The ranking model learns to examine each combination of the candidate prediction and its retrieved events—or, in other words, its *evidence*—and assign high scores to the candidates that are strongly supported by the retrieved evidence. Section 3.3 discusses this phase.

## 3.1 Phase-I: Proposing Predictions

Given a history of previous events $\mathcal{H}_i = (t_1, k_1), \ldots, (t_{i-1}, k_{i-1})$, the base event sequence model is used as a proposer to generate candidate predictions on the time and type of the next event.

For time prediction, we draw $L$ i.i.d. samples $\hat{t}_i^{(1)}, \ldots, \hat{t}_i^{(L)}$ from the base model via the thinning algorithm. If we were to only use this base model but not our LLM-enhanced framework, the final MBR time prediction would be the average of the samples, i.e., $\hat{t}_i = \frac{1}{L} \sum_{\ell=1}^{L} \hat{t}_i^{(\ell)}$. However, the MBR prediction may not be accurate since the base model is imperfect. Therefore, our LAMP framework treats all the $L + 1$ samples—with $\hat{t}_i^{(L+1)}$ denoting the MBR prediction—as candidates, and utilize the LLM and ranking model to score them in later phases. If any of the $L$ draws is actually a better prediction than the MBR estimate, our framework has a chance to rank it higher.

As we'll show shortly in sections 3.2 and 3.3, the LLM and ranking model work on full events. So we find the most probable $M$ full events $\{(\hat{t}_i^{(\ell)}, \hat{k}_i^{(\ell,m)})\}_{m=1}^{M}$ for each time proposoal $\hat{t}_i^{(\ell)}$, where $\hat{k}_i^{(\ell,m)}$ is the event type that has the $m$-th highest intensity at time $\hat{t}_i^{(\ell)}$.

For type prediction given the ground-truth time $t_i$, we find $M$ event types $\hat{k}_i^{(1)}, \ldots, \hat{k}_i^{(M)}$ where $\hat{k}_i^{(m)}$ has the $m$-th highest intensity at time $t_i$ under the base model. If we were to only use this base model for prediction, the MBR type prediction would be the event type $\hat{k}_i^{(1)}$ with the highest model intensity. However, our full framework will use the LLM and ranking model to examine each of the top $M$ full events $\{(t_i, \hat{k}_i^{(m)})\}_{m=1}^{M}$ at time $t_i$ in order to make a more informed prediction.

In practice, the event types often have structures and we may be interested in predicting an attribute of the structured type. Figure 1 shows an example in which we are trying to predict the predicate of the structured event type given its time, subject, and object. In such cases, we just need to select as proposals the most probable $M$ event types whose other attributes are the same as the known information (e.g., Tesla and Australia in Figure 1).

```
I want you to do the reasoning over          ## Example 1
social events. I given you an effect
event and you give me four or five           effect
cause events. An effect event is an          predicate: APPEAL
event that happens. A cause event is         time: 2022-04-23
believed to be one of the causes that        subject: GERMANY
have triggerred the effect event to          object: GREEN PROJECT
happen. Each event consists of a time,
a type (that includes subject,               reasoning:
predicate, object), and a news headline      -----------------------
 describing the event.                        cause event 1
                                             predicate: REDUCE RELATIONS
The predicates are restricted to the 20      time: 2022-04-21
 options below.                              subject: EUROPE
1. MAKE STATEMENT                            object: RUSSIA
   .                                         headline: Europe determined to ban
   .  // Full list are in Appendix E.4.      Russian energy exports.
   .
20. ENGAGE IN MASS VIOLENCE                   cause event 2
                                             predicate: DISAPPROVE
Now I give you 10 examples. In each          time: 2022-03-16
example, the first event is the effect       subject: EUROPE
and the next several events are the          object: RUSSIAN
causes that happened earlier.                 headline: Europe can endure painful
   .                                         transition to live without Russian oil.
   .  // Examples are in Listing 2.
   .                                            .
Now please generate possible causes for         .  // Other causes are in Appendix E.4.
                                                .
effect
predicate: CONSULT                            ## Example 2
time: 2022-07-05                                .
subject: CHINA PM                               .  // Other examples in Appendix E.4.
object: YELLEN
```

Listing 1: Format of our LLM prompt.                Listing 2: Few-shot examples in our prompt.

The $L$ and $M$ are hyperparameters. Ideally, we would like to analyze all the possible candidates (i.e., $L = \infty$ and $M = |\mathcal{K}|$), which is intractable for time prediction and expensive for type prediction. In practice, we focus on the most plausible candidates to maintain a low computation cost. In our experiments, our framework already performs remarkably well with small $L$ and $M$.

### 3.2 Phase-II: Prompting LLM to Perform Abductive Reasoning

For each proposed event $(t, k)$, our framework selects a set of previous events from its full history as its supporting evidence $\mathbf{e}(t, k)$. The selection is guided by an LLM (e.g., GPT-3.5). Technically, we prompt the LLM to imagine some possible cause events that—under the LLM's belief—would be able to explain the occurrence of this proposal. The imaginary cause events may not exactly match any actual event in the history, but we could use them as queries to search for the most similar ones.

Prompting is a widely used technique to extract knowledge from an LLM. A prompt is a concise statement designed to elicit a response from the LLM. It typically includes the task description and a few demonstrations. Our prompt follows the format in Listings 1 and 2.

Each LLM-generated cause event is used as a query to search for $D$ most similar events in the history, where $D > 0$ is a hyperparameter. The overall evidence $\mathbf{e}(t, k)$ is then defined to be the union of the previous events retrieved by the LLM-generated causes. Retrieval is vector-based: we construct a query embedding $\mathbf{v}_q$ for the query event, and a key embedding $\mathbf{v}_k$ for each actual previous event; the similarity is measured by cosine between $\mathbf{v}_q$ and $\mathbf{v}_k$, i.e., $\frac{\mathbf{v}_q^\top \mathbf{v}_k}{\|\mathbf{v}_q\|\|\mathbf{v}_k\|}$. We use a pretrained SentenceBERT (SBERT) (Reimers & Gurevych, 2019) as our embedding model. The model takes as input a text string concatenating the event time $t$, the text identifier (e.g., subject-predicate-object) of the event type $k$, and—if any—the textual mark $\mathbf{m}$ of the event. It returns an embedding for each token, and we take the event embedding $\mathbf{v}$ to be the average of the SBERT embeddings.

### 3.3 Phase-III: Ranking Proposals

In this phase, our framework scores each proposed event $(t, k)$ based on the compatibility with its retrieved evidence $\mathbf{e}(t, k)$. Precisely, the score is defined to be

$$s_{\text{event}}(t, k) \stackrel{\text{def}}{=} \exp\left(c\left((t, k), \mathbf{e}(t, k)\right)\right) \tag{4}$$

The function $c$ takes as input the proposed event $(t, k)$ as well as its evidence $\mathbf{e}(t, k)$, and returns a scalar $\in \mathbb{R}$. A high value of $c$ means that this proposal is strongly supported by its retrieved evidence, and thus is more likely to be an actual event at time $t$; a low value means that this proposal has no strong evidence even after we have tried our best to search from the history.

Given the most probable $M$ events $\{(t, k^{(m)})\}_{m=1}^{M}$ at time $t$, we sum their $s_{\text{event}}$ scores to measure the overall belief of our framework in an event occurring at time $t$. That is,

$$s_{\text{time}}(t) \stackrel{\text{def}}{=} \sum_{m=1}^{M} s_{\text{event}}(t, k^{(m)}) \tag{5}$$

Intuitively, this score is high when any of the top-ranked event types at this time can be strongly supported by the retrieved evidence. Otherwise, even the top-ranked event types have no strong evidence in the history, which implies that the next event is unlikely to occur at this time.

For time prediction, each proposed time $\hat{t}^{(\ell)}$ $(\ell = 1, \ldots, L+1)$ has a score $s_{\text{time}}(\hat{t}^{(\ell)})$—more precisely, $\sum_{m=1}^{M} s_{\text{event}}(\hat{t}^{(\ell)}, \hat{k}^{(\ell,m)})$—and our final prediction is the proposal with the highest score.

For type prediction given time $t$, each proposed type $\hat{k}^{(m)}$ $(m = 1, \ldots, M)$ has a score $s_{\text{event}}(t, \hat{k}^{(m)})$, and our framework takes the final prediction to be the type with the highest score.

**Model architecture.** Our function $c$ is an energy function with a continuous-time Transformer architecture (Xue et al., 2022). It reads the proposal $(t, k)$ followed by its evidence events in the chronological order, and returns a compatibility score $\in \mathbb{R}$.

We choose this architecture because its continuous-time attention is suitable for our setting. First, the attention mechanism may learn to disregard any of the retrieved events that are not really relevant and focus on those which can actually support the proposal. Second, its sophisticated handling of time may capture how the time of an evidence event may influence its relevance to the proposal (e.g., a recent evidence event may be more important than an ancient event).

**Training.** We train the ranking model by maximizing the objective $J \stackrel{\text{def}}{=} J_{\text{actual}} + \beta J_{\text{no}}$ where $\beta \geq 0$ is a hyperparameter. The first term $J_{\text{actual}}$ is defined to be

$$J_{\text{actual}} \stackrel{\text{def}}{=} \sum_{i=1}^{I} \left( \log s_{\text{event}}(t_i, k_i) - \log \left( s_{\text{event}}(t_i, k_i) + \sum_{m=1}^{M} s_{\text{event}}(t_i, k_i^{(m)}) \right) \right) \tag{6}$$

where $(t_1, k_1), \ldots, (t_I, k_I)$ is a sequence of events over the time interval $(0, T)$, and each $k_i^{(m)}$ is the event type with the $m$-th highest intensity under the base model at time $t_i$. By maximizing $J_{\text{actual}}$, the function $c$ learns to increase the scores of the events that have actually happened, but suppress the scores of the non-events at times $t_1, \ldots, t_I$. The second term $J_{\text{no}}$ is defined to be

$$J_{\text{no}} \stackrel{\text{def}}{=} -\sum_{n=1}^{N} \log s_{\text{time}}(t_n) = -\sum_{n=1}^{N} \log \sum_{m=1}^{M} s_{\text{event}}(t_n, k_n^{(m)}) \tag{7}$$

where each $t_n$ is a time point uniformly sampled from $(0, T)$ and each $k_n^{(m)}$ is the event type with the $m$-th highest intensity at time $t_n$. By maximizing $J_{\text{no}}$, the function $c$ learns to decrease the scores of the non-events at times over $(0, T)$ other than $t_1, \ldots, t_I$.

Although not explicitly mentioned in equations (6) and (7), computing $J_{\text{actual}}$ and $J_{\text{no}}$ involves searching for the evidence of the actual and proposed events.

## 4 Experiments

Our code is at `https://github.com/iLampard/lamp`. This repository includes CSV files containing numerical results of our experiments. It also includes qualitative results such as LLM-generated cause events. Experiment details (e.g., hyperparameters) are in Appendix E.

## 4.1 Experimental Setup

We conducted experiments on three real-world datasets (see Appendix E.1 for dataset details).

**GDELT** (Leetaru & Schrodt, 2013). The GDELT Project monitors events all over the world, with live datasets updated every 15 minutes. We only focused on the political events that happened in G20 countries from 2022-01-01 to 2022-07-31, ending up with a corpus of 109000 time-stamped event tokens. This choice of time range guarantees that our data was not included in the training data of the most recent GPT. The event type $k$ of each token has a structured name of the format subject-predicate-object. Each predicate is one of the twenty CAMEO codes such as CONSULT and INVESTIGATE (see Appendix E.4 for a full list); each subject or object is one of the 2279 political entities (individuals, groups, and states) such as Tesla and Australia. So there are about 104M event types in total, making this dataset extremely challenging. Each event token has a news headline that concisely describes the event. We split the dataset into disjoint train, dev, and test sets based on their dates: the 83100 events that happened before 2022-07-05 are training data; the 16650 events after 2022-07-19 are test data; the 9250 events between these dates are development data.

**ICEWS** (Boschee et al., 2015). Similar to GDELT, this dataset logs interactions between social-political entities. We collected 79410 event tokens from 2022-10-11 to 2023-02-28. Its event types have the same structure as GDELT: each predicate is one of the twenty CAMEO codes; each subject or object is one of the 2981 political entities. We split the dataset into disjoint train, dev, and test sets based on their dates: the 41600 events that happened before 2023-01-16 are training data; the 22030 events after 2023-02-01 are test data; the 15780 events between these dates are development data.

**Amazon Review** (Jianmo Ni, 2019). This dataset contains user reviews on Amazon shopping website from 2014-01-04 to 2016-10-02. We focused on the most active 2500 users and each user has a sequence of product review events. The type $k$ is the category of the product: we selected the most frequently-reviewed 23 categories and grouped all the others into a special OTHER category, ending up with 24 categories in total. Each review event also has a mark $\mathbf{m}$ which is the actual content of the review. Each of the 2500 sequences is cut into three segments: the events that happened before 2015-08-01 are training data; those after 2016-02-01 are test data; the events between these dates are dev data. Then we have 49,680 training tokens, 7,020 dev tokens, and 13,090 test tokens.

We experimented with four state-of-the-art event sequence models: NHP (Mei & Eisner, 2017a), Know-Evovle (KE) (Trivedi et al., 2017), DyRep (Trivedi et al., 2019), and ANHP (Yang et al., 2022). For each of them, we evaluated it as a baseline method as well as integrated it into our LAMP framework. KE and DyRep require domain-specific knowledge to configure their structural-sparse architectures: we evaluated them on GDELT since their GDELT-specific architectures are available in the original papers; we didn't evaluate them on Amazon Review since we do not have such knowledge on this data. ANHP can take domain knowledge into its architecture but it is optional, so we evaluated it on both GDELT and Amazon data: on GDELT, we adapt the knowledge used in KE and DyRep into its structure; on Amazon Review, we use the generic architecture. On Amazon Review, we also experimented with NHP since it doesn't require any domain-specific structure knowledge.

We experimented with three strong LLMs: GPT-3-davinci (Brown et al., 2020) which we also denote as G3.0; GPT-3.5-turbo (Brown et al., 2020; Stiennon et al., 2020; Gao et al., 2022) which we also denote as G3.5, and Llama-2-chat with 13B parameters (Touvron et al., 2023) which we also denote as llama. For GDELT and ICEWS data, we used 10-shot prompts; for Amazon Review data, we used 8-shot prompts. Each "shot" is a demonstration that contains an effect event followed by one or more expert-annotated cause events. Prompt examples can be found in Appendix E.4.

## 4.2 Main Results on Type and Time Prediction

Our main results are displayed in Figure 2. Figure 2a shows the result of each method on GDELT data. GDELT data is updated every fifteen minutes so the time intervals are regular and thus it is not interesting to predict them. For type prediction, we focus on predicting certain attributes given the others, which is more practical than full type prediction. In practice, predicting "which of the hundreds of millions of events is the most probable" is too difficult and existing models will all perform disastrously. But answering questions like "what will A do to B" and "to whom A will do this" is usually useful enough for real applications. Note that attribute prediction is still very challenging: e.g., there are 45580 distinct predicate-object combinations in GDELT data.

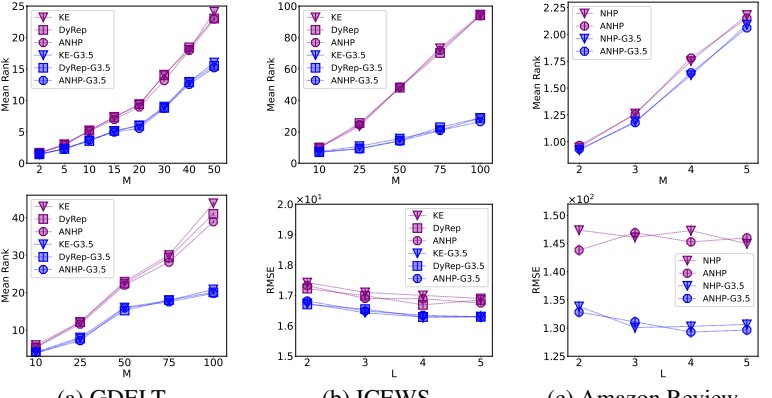

| (a) GDELT. | (b) ICEWS. | (c) Amazon Review. |

Figure 2: Prediction performance of different methods on each dataset. On GDELT, the upper figure is for object prediction, and the lower figure is for predicate-object joint prediction. On ICEWS, the upper figure is for object prediction, and the lower figure is for time prediction. On Amazon Review, the upper figure is for type prediction, and the lower figure is for time prediction.

We evaluate each model by the quality of its top-ranked predictions. For each baseline model (KE, DyRep, or ANHP), the list of top predictions contains the top $M$ event types (with known attributes filled, if any) that have the highest intensities; see section 3.1. Our LLM-enhanced framework takes the list given by its base model, and sorts it based on the $s_{\text{event}}$ scores of the proposals. Our primary evaluation metric is the mean rank (MR). This metric has a straightforward interpretation: given a sorted list of proposed predictions, it measures the average rank of the ground-truth type in the list; a smaller MR means a higher rank, and thus a better result (e.g., MR = 1 means "ranked at the top on average"). We also used the mean reciprocal rank (MRR), which is less interpretable but more robust to bad predictions than MR. Appendix E.2 includes full procedures for computing these metrics. In our experiments, MR and MRR results yield the same conclusions. So we present MR results in this section for its straightforward interpretation, but leave MRR results to Appendix F.

On ICEWS and Amazon Review, we evaluate each model on time prediction (in addition to type prediction), which is measured by the root of mean squared error (RMSE). For each held-out token, each base model proposes a list of scored predictions; see section 3.1. Our LAMP framework reranks the list given by its base model. In either case, the final prediction is the highest-ranked proposal.

In each evaluation setting shown in Figure 2, our LLM-enhanced framework substantially and consistently outperforms the corresponding baseline model across a range of $L$ and $M$ values. All the results throughout the paper have 95% bootstrap confidence intervals but they are too tight to be visually obvious in most cases, implying that our improvements are significant. When we draw more proposals from the base model (i.e., $L$ and $M$ are larger), our framework tends to enjoy a larger improvement over the base model. For predicate-object joint prediction on GDELT, the ground-truth type ranks at around 40 in the lists given by the base models, but is moved to around 20 by our LLM-based adjustment. For object prediction on ICEWS, our method improved the MR results from around 90 to about 20. Note that it is not fair to compare the same method across $L$ or $M$.

The MR and MRR are only evaluated on the held-out tokens whose ground-truth event types fall into the top proposals. But how many are there such tokens? This solely depends on how well the base event sequence model works. A detailed analysis can be found in Appendix F.2.

### 4.3 Analysis

Now we present our analysis on GDELT. On GDELT, we focus on the predicate and object prediction. It is less expensive than the time and predicate-object prediction (which requires an order of magnitude more GPT hours and GPT API calls) but the results are well correlated in our pilot experiments. More analysis can be found in Appendix F, including analysis on the other datasets.

**Analysis-I: About LLMs.** Section 4.2 only shows the results of the GPT-3.5 version of our framework. Figure 3 shows the results of the other versions with ANHP as the base model. These LLMs all help improve the prediction performance but GPT-3.5 and Llama work significantly better than GPT-3. Interestingly, the Llama version performs competitive to the GPT-3.5 version. Its strong performance demonstrates that our framework doesn't have to depend on black-box LLMs; indeed, it can excel with open-source LLMs like Llama. Note that Llama has considerably fewer parameters than GPTs. An important reason for its success may be the reinforcement learning from human

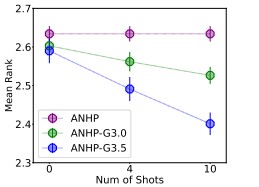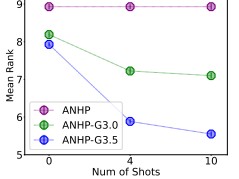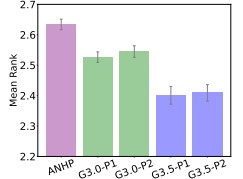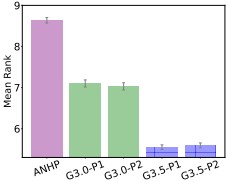

Figure 3: MR of using different LLMs on predicate (left) and object (right) prediction on GDELT.

Figure 4: Effect of the number of evidence events on predicate (left) and object (right) prediction on GDELT.

Figure 5: Effect of the number of shots: MR of predicate (left) and object (right) prediction on GDELT.

Figure 6: Effect of the shot choices: MR of predicate (left) and object (right) prediction on GDELT.

feedback (RLHF) technique, which is adopted by GPT-3.5 and Llama-2-chat, but not by GPT-3. Further investigation may be an interesting direction for future work.

**Analysis-II: How many evidence events do we need?** In our GDELT experiments, we varied the number of retrieved evidence events for each proposed prediction. As shown in Figure 4, the results improve when the number of retrievals increases from 5 to 10, but become worse when the number further goes up. Apparently, if we retrieve too many "evidence" events, then most of them might not be actually helpful so the performance will degrade due to the decreased signal-to-noise ratio.[1] We also tried another retrieval criteria: instead of retrieving a fixed number of previous events, we only retrieve an event if its similarity score exceeds a prespecified threshold. It turns out that this criteria only had negligible impact on the results.

**Analysis-III: Prompt design.** How will the design of the prompt influence the performance of our LAMP framework? As shown in Figure 5, LAMP performs better when the prompt includes a larger number of demonstrations, and the GPT-3.5 version consistently outperforms the GPT-3 version. Interestingly, even 0-shot prompting yields a better result than the baseline ANHP model, emphasizing that the LLMs are indeed very helpful.

Figure 6 shows how sensitive our framework is to the choice of demonstrations. We use P1 to denote the set of demonstrations used throughout our main experiments. Then P2 refers to a new set of demonstrations obtained by randomly replacing half of the demonstrations in P1. As we can see, changing the demonstrations has only a slight effect on performance, and LAMP consistently outperforms ANHP, whether using P1 or P2.

We designed the prompt templates without prior experimentation or tuning on any dataset. In a post analysis, we tested several different templates on a small subset of GDELT data, and found only minimal variations in results, as long as the task description was clear. Interestingly, a new template that we tested doesn't include the full list of possible predicates (see Listing 1), yet the results with this template closely match those of our original version.

**Analysis-IV: About generalization.** The LLM in LAMP is instructed by a few demonstrations. On GDELT, these demonstrations cover 17 out of 20 (85%) predicates and 30 out of 2279 (1.31%) entities. However, GPT-3.5 is able generalize beyond the demonstrations, suggesting cause events that involve novel predicates and entities. Precisely, the LLM-generated cause events involve 4699 distinct predicates and 65554 distinct entities, covering all the 20 ground-truth predicates and more than 800 ground-truth entities. Appendix F.3 includes more details about this analysis.

LLMs are known for strong in-context learning capability. In our specific problem setting, an LLM may have already understood the meanings of the predicates and been familiar with many entities

---

[1]For Figures 4–6, we fixed $M = 5$ for predicate prediction and $M = 20$ for object prediction.

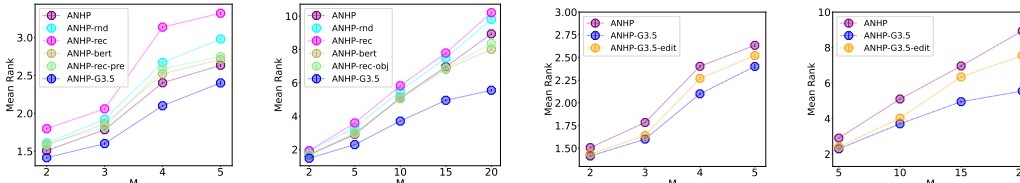

Figure 7: Predicate (left) and object (right) prediction of different retrieval methods on GDELT.

Figure 8: Effect of similarity metric: MR of predicate (left) and object (right) prediction on GDELT.

through its pretraining and fine-tuning processes. Consequently, when presented with a proposal, it can follow demonstrations and draw upon its internal knowledge to suggest plausible causes. Benefiting from the strong generalization capability of LLMs, our LAMP framework has a significant potential for broad applications.

**Analysis-V: About retrieval methods.**   To further investigate the usefulness of LLMs, we tested several versions of our framework that do not involve the LLM in retrieval: rnd randomly samples 10 previous events from the history for each proposal; rec uses the most recent 10 past events; rec-pre retrieves the most recent 10 events that share the same predicate with the proposal; rec-obj retrieves the most recent 10 events that share the same object; bert uses SBERT to embed the text-based identifier of each event and retrieve the most similar 10 events based on the cosine similarity of SBERT embeddings. As shown in Figure 7, all these retrieval methods perform significantly worse than our LLM-based method. Noticeably, the "most recent" retrievals work poorly. It suggests that "recent" is not a good inductive bias for our problem setting. On dev data, we found that the cause events retrieved based on LLM-generated clues are often not "recent".

We also tested using the edit distance as the similarity score in our LLM-based retrieval method. For a pair of events, we compute the character-level edit distance (Jurafsky & Martin, 2009) between their text strings that are otherwise fed into the SBERT embedding model (section 3.2), and define the similarity to be the reciprocal of this distance. As shown in Figure 8, using this metric yields a better result than the baseline method, but performs worse than our original framework.

**Analysis-VI: About data leakage.**   GPT models are trained on online text data up to 2021.[2] This dataset doesn't include the GDELT or ICEWS data used in our experiments. Our demonstrations span a wide time range in training data, and there is a large temporal gap between the training and test data. Precisely, the most recent demonstration is about an event on 2022-06-07, while the earliest test event occurred on 2022-07-05. Below are the percentiles for the time differences (in days) between the test events and the latest demonstration:

| percentage | 1% | 5% | 25% | 50% | 75% | 95% | 99% |
|---|---|---|---|---|---|---|---|
| percentile | 40 | 41 | 44 | 52 | 54 | 55 | 55 |

The GPT training data may have covered the Amazon Review data. Therefore, we had a conversation with GPT-3.5, checking whether it could recall information about specific users or specific reviews. GPT-3.5 could say nothing specific about the users or reviews, indicating that we do not have an issue of data leakage. Our conversation can be found in Appendix E.5.

## 5   Discussion and Related Work

Event reasoning and prediction is an important problem that arises in various real-world applications. A great number of event sequence models have been proposed and advanced this field, including the classical graphical models (Hawkes, 1971; Du et al., 2015a), recurrent neural models (Du et al., 2016; Mei & Eisner, 2017b; Xiao et al., 2017; Omi et al., 2019; Shchur et al., 2020; Boyd et al., 2020), and Transformer-based models (Zuo et al., 2021; Zhang et al., 2020; Enguehard et al., 2020; Sharma et al., 2021; Zhu et al., 2021; Yang et al., 2022; Xue et al., 2022; Liu et al., 2022). These models have been applied to a wide range of applications such as network analysis (Choi et al., 2015; Etesami

---

[2]GPT-3.0-davinci's training data goes up to October 2019, while GPT-3.5-turbo's training data goes up to September 2021 (source: `https://platform.openai.com/docs/models/gpt-3-5`).

et al., 2016), recommendation systems (Du et al., 2015b), social analysis (Guo et al., 2015; Lukasik et al., 2016; Zhang et al., 2022), and healthcare (Hua et al., 2022; Zhang et al., 2023b).

Recently, there has been a growing body of research that has directed its focus towards the textual features of the real-world events, such as the reports and news articles about the events. This line of work includes Deng et al. (2020, 2021, 2022); Han & Ning (2022). Their methods all treat text as auxiliary features but do not consider reasoning about that text information. Our work is significantly different from this line of research since we focus on reasoning. Our framework reasons about the events and their text information (e.g., text identifiers of the event types, associated documents) by prompting a large language model. The large language model has consumed a massive amount of text during pretraining and is aware of diverse world knowledge, thus able to perform the kind of reasoning beyond the capacities of the aforementioned models.

By leveraging large language models, our framework induces domain-specific knowledge into the deployment of event sequence models. It is related to previous work of configuring domain-specific knowledge into model architectures (Trivedi et al., 2017, 2019; Mei et al., 2020; Yang et al., 2022). But our work is significantly different: it doesn't rely on human domain experts to write down a full set of knowledge; instead, we extract knowledge from language models via few-shot prompting. Prompting has become a standard way of applying large language models to reasoning tasks. There has been a recent surge of work that develops novel prompting techniques for a better performance (Wei et al., 2022a,b; Zhou et al., 2023). Our framework is general and can adopt any prompting methods.

This paper is closely related to research in logical reasoning, which primarily focuses on proving goals from known facts and rules. A major approach for this problem is backward chaining: it works backward from the goal, chaining through rules to find known facts that support the proof (Russell & Norvig, 2010). This approach has been applied to various application problems, including reasoning and planning in natural language (Ye et al., 2022; Weir & Van Durme, 2022; Kazemi et al., 2023; Kassner et al., 2023). In our setting, each proposed event is treated as a goal, and previous events serve as known facts. Like backward chaining, our method identifies previous events that support the proposal. But our method only performs a single step of reasoning, assuming complete data where all events are observable. A second step of reasoning is unnecessary since direct causes are observable and more temporally recent than indirect causes. In cases of incomplete data, multiple reasoning steps may be required to identify indirect causes. Handling incomplete data will be a non-trivial extension of our current framework, which we leave to future research.

This paper is also closely related to research in event-centric natural language processing (NLP) (Chen et al., 2021). Over the past decades, there has been a great amount of progress in this area, including many datasets and benchmarks (Ning et al., 2020; Li et al., 2020; Han et al., 2021; Wen et al., 2021; Feng et al., 2022) as well as a diversity of methods for key problems such as event detection and extraction (Ji & Grishman, 2008; Li et al., 2013; Feng et al., 2016; Lyu et al., 2021; Wang et al., 2021), relation extraction and prediction (Chan & Roth, 2011; Ning et al., 2018b; Wang et al., 2020; Wen & Ji, 2021; Li et al., 2022), event induction and summarization (Do et al., 2012; Saravanakumar et al., 2021; Li et al., 2021b,a; Jin et al., 2022), and temporal reasoning (Ning et al., 2018a; Ballesteros et al., 2020; Zhou et al., 2020, 2021). Recently, there has been research in leveraging LLMs to solve these problems (Dror et al., 2023; Li et al., 2023; Zhang et al., 2023a). Our work complements this line of research, focusing on the problem of event sequence modeling, which has been out of the scope of the classical event-centric NLP.

## 6   Conclusion

In this paper, we present LAMP, a general modeling and prediction framework that leverages the abductive reasoning ability of large language models to help improve the prediction performance of event sequence models. Empirical studies demonstrate that our LLM-enhanced framework can significantly outperform the state-of-the-art event sequence models. Our findings have significant implications for future research in the field of event sequence modeling. In principle, an event sequence model should benefit from a range of reasoning abilities of large language models such as deductive reasoning, inductive reasoning, commonsense reasoning, and arithmetic reasoning. Further exploration in this area may lead to impactful innovations.

## Acknowledgments

This work was supported by a research gift to the last author by Adobe Research. We thank the anonymous NeurIPS reviewers and meta-reviewer for their constructive feedback. We also thank our colleagues at Ant Group, UChicago, and TTIC for helpful discussion.

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

# Appendices

## A  Societal Impacts

Our paper develops a novel method to integrate large language models into temporal modeling. By describing the model and releasing code, we hope to facilitate the modeling of continuous-time sequential data in many domains. However, like many other machine learning models, our method may be applied to unethical ends. For example, its abilities of better fitting data and making more accurate predictions could potentially be used for unwanted tracking of individual behavior, e.g. for surveillance.

## B  Limitations

Our framework utilizes LLMs such as GPT-3.5 and thus inherits the issues of those models such as hallucination and biased content. Therefore, the cause events generated by LLMs may be irrelevant, incorrect, or misleading, thus harming the overall performance of our framework. Also because of using LLMs, our approach requires human-crafted prompts, and the events need to have textual information (e.g., event types have textual identifiers or marks).

Our framework needs a pretrained event sequence model, and its overall capability is partially determined by this base model. When the base model can handle events with duration (i.e., events that last for a period of time, such as holding a tool), our framework will inherit this merit. But our framework will also inherit some of the technical limitations of its base event sequence model. If the base model can only handle a finite set of event types, then our framework will inherit this limitation. If the base model is misspecified or poorly trained, our framework may not work well.

Our compatibility function $c$ is a Transformer-based neural network which is known to be data-hungry. Though it worked well in our experiments, it might struggle when starved of data.

## C  Possible Extensions

Our framework currently trains each component separately. A natural improvement will be to jointly train its three components. In principle, we can iteratively

- refine the base event sequence model with the feedback from the LLM and ranking model;
- learn to prompt the LLM so that it generates cause events of higher quality;
- train the ranking model with proposals and evidence given by the improved base model and LLM.

Another interesting extension of our framework is to apply its propose-justify-rank pipeline to other problems. At the core of our framework is the LLM, which is able to examine each proposed prediction drawn from the base model, and suggest clues to justify its validity. This idea seems to be a natural fit to a wide range of reasoning problems. Question answering is an example: given a question, an existing model proposes multiple plausible answers; an LLM reads each proposed answer and reasons about its preconditions; a search module finds out evidence (from local knowledge bases or the open internet) which may match the LLM-suggested preconditions; a ranking model learns to score each combination of answer and evidence.

## D  Method Details

### D.1  ANHP

On Amazon data, the ANHP model we used is directly from Yang et al. (2022). On GDELT and ICEWS data, we modified the definition of event embedding so that it is easier to scale up to hundreds of millions of event types. Particularly, ANHP defines a dynamic event embedding $[\![k]\!](t)$ for the event type $k$ at time $t$. On GDELT and ICEWS, each event type is a tuple of subject-predicate-object, so we redefine the event embedding to be

$$[\![k]\!](t) \stackrel{\text{def}}{=} \text{concat}\left([\![k^{\text{subject}}]\!](t), [\![k^{\text{predicate}}]\!](t), [\![k^{\text{object}}]\!](t)\right) \tag{8}$$

and keep everything else the same as in the original ANHP.

This modified version is more scalable since it needs to maintain a significantly smaller number of dynamic embeddings: there are hundreds of millions of possible event types, but there are only thousands of possible subjects and objects.

# E    Experimental Details

## E.1    Dataset Details

Table 1 shows statistics about each dataset in our experiments.

| DATASET | # OF ENTITIES | # OF PREDICATES | # OF EVENT TOKENS | | |
|---------|---------|---------|-------|-----|------|
| | | | TRAIN | DEV | TEST |
| GDELT | 2279 | 20 | 83100 | 9250 | 16650 |
| ICEWS | 2981 | 20 | 41600 | 15780 | 22030 |
| AMAZON | 1 | 24 | 49680 | 7020 | 13090 |

Table 1: Statistics of datasets.

## E.2    Evaluation Metric Details

Now we give the formal definitions of MR and MRR. The primary evaluation metric is the mean rank (MR): for each method, we initialize $N = R = 0$; for each held-out token, the method provides a list of $M$ proposed predictions sorted in the decreasing order of their scores; if the ground-truth type is among the proposals, we update $N \mathrel{+}= 1$ and $R \mathrel{+}= r$ where $r$ is the rank of the ground-truth type in this list; the mean rank is $R/N$ and the lower is the better.

The mean reciprocal rank (MRR) is as follows: for each method, we initialize $N = R = 0$; for each held-out token, the method provides a list of $M$ proposals sorted in the decreasing order of their scores; if the ground-truth type is among the proposals, we update $N \mathrel{+}= 1$ and $R \mathrel{+}= 1/r$ where $r$ is the rank of the ground-truth type in this list; the MRR is $R/N$ and the higher is the better.

Note that MR and MRR are not affected by the fact that there might be multiple ground-truth attributes. That is because the contribution of $j^{\text{th}}$ ground-truth (among the multiple) to the final number is only dependent on its rank in the list but not its $j$ index.

Next we give the precise definitions of MAP and MAR.

MAP is computed as follows. We first initialize $N = C = 0$. Then, for each partial held-out event (i.e., only $t$ given in Amazon Review, $t$ and some attributes given in GDELT), the model gives its top $M$ predictions on the attribute of interest (e.g., type $k$ on Amazon Review and "object" on GDELT). There might be multiple ground-truth attributes for a held-out event since multiple events may be recorded at the same time (due to time quantization and recording errors). If any of the ground-truth is in the top $M$ list, we update $N$ and $C$: first, we have $N \mathrel{+}= J$ where $J \leq M$ is the number of the ground-truth attributes that are covered in the top $M$ list; then, for $j^{\text{th}}$ covered ground-truth, we let $C \mathrel{+}= j/R$ where $R$ is its rank in the top $M$ list. In the end, MAP is defined as $C/N$. Each $i/R$ is a pseudo-count for this event: it is in $(0, 1]$ since the rank $R$ of the $j^{\text{th}}$ correct prediction will be surely $\geq j$; if it ranks at the top, $i/R$ is close to 1 and it is counted as "predicted correctly"; if it ranks at the bottom, $i/R$ is close to 0, meaning that it is almost missed by the top $M$ proposals.

MAR is defined as follows. We first initialize $N = C = 0$. For each partial held-out event, the model gives its top $M$ proposals on the attribute of interest; for the $j^{\text{th}}$ correct prediction in the list, we let $C \mathrel{+}= j/R$ where $R$ is its rank in the top $M$ list; in the end, MAR is defined as $C/N$ where $N$ is the number of actual events.

## E.3    Implementation Details

All models are implemented using the PyTorch framework (Paszke et al., 2017).

For the implementation of NHP, AttNHP and energy functions, we used the code from the public Github repository at `https://github.com/ant-research/EasyTemporalPointProcess` (Xue et al., 2023) with Apache License 2.0.

For the implementation of Know-Evolve, we used the code from the public Github repository at `https://github.com/rstriv/Know-Evolve` (Trivedi et al., 2017) without any license.

For the implementation of DyRep, we used the code from the public Github repository at `https://github.com/uoguelph-mlrg/LDG` (Trivedi et al., 2019) without any license.

For the implementation of MAR@M and MAP@M, we used the code from the public Github repository at `https://github.com/statisticianinstilettos/recmetrics` without any license.

For the implementation of Levenshtein distance, we used the code from the public Github repository at `https://github.com/maxbachmann/Levenshtein.git` with General Public License 2.0.

To compute the dense representations of text descriptions of events, we use the code from the public Github repository at `https://github.com/UKPLab/sentence-transformers` with Apache License 2.0.

### E.4 Prompts

We show one example of the prompt structures used for GDELT dataset as below. The effect event consists of an event predicate, an event time, a subject name and an object name while the cause events consists of an event headline additionally.

```
I want you to do the reasoning over social events. I given you an effect event
and you give me four or five cause events. An effect event is an event that
happens. A cause event is an event that is believed to be one of the causes that
trigger an effect event to happen. Each event consists of an event headline, an
event predicate, an event time, subject name and object name of describing the
event.

The predicates of the effect and cause events are restricted to 20 options, with
names (in capital) and the descriptions below.
1. MAKE STATEMENT: decline comment, make comments.
2. APPEAL: appeal for material, economic, military, humanitarian aid or
cooperation.
3. EXPRESS INTENT TO COOPERATE: Express intent to engage in material, diplomatic,
 military aid.
4. CONSULT: make a visit, host a visit.
5. ENGAGE IN DIPLOMATIC COOPERATION: praise or endorse, defend verbally.
6. ENGAGE IN MATERIAL COOPERATION: cooperate economically, militarily, judicially.

7. PROVIDE AID: provide economic, military, humanitarian aid.
8. YIELD: ease admin or political sanctions or military blockade, return and
release.
9. INVESTIGATE: investigate crime, corruption, human rights abuses, military
actions.
10. DEMAND: demand any type of cooperation, aid, reforms, rights, easing of
sanctions.
11. DISAPPROVE: criticize or denounce, accuse of crime, human rights abuses,
complain officially and make lawsuit against.
12. REJECT: reject material, economic, military, judicial operations, requests or
 plans.
13. THREATEN: threaten to reduce aid, to boycott, to reduce or break relations,
to impose sanctions.
14. PROTEST: civilian demonstrations.
15. EXHIBIT MILITARY POSTURE.
16. REDUCE RELATIONS: reduce or break any relations.
17. COERCE: seize or damage properties, impose administrative sanctions or
restrictions.
18. ASSAULT: use of unconventional forms of violence.
19. FIGHT: uses of conventional force.
20. ENGAGE IN UNCONVENTIONAL MASS VIOLENCE.

Now I give you 10 examples of event reasoning. In each example, the first event
is the effect event, the next three to five events are cause events that happen
earlier.
```

```
## Example 1

effect
predicate: APPEAL
time: 2022-04-23
subject: GERMANY
object: GREEN PROJECT

reasoning:
--------------------------
cause event 1
predicate: REDUCE RELATIONS
time: 2022-04-21
subject: EUROPE
object: RUSSIA
headline: Europe determined to ban Russian energy exports.

cause event 2
predicate: DISAPPROVE
time: 2022-03-16
subject: EUROPE
object: RUSSIAN
headline: Europe can endure painful transition to live without Russian oil.

cause event 3
predicate: ENGAGE IN DIPLOMATIC COOPERATION
time: 2022-03-05
subject: BUSINESS
object: GOVERNMENT
headline: Oil prices surge to multi-year highs as Ukraine conflict causes ripple
effect in global oil supplies.

cause event 4
predicate: REJECT
time: 2022-03-04
subject: SENATOR
object: RUSSIA
headline: Marshall, Moran seek ban on Russian oil imports, urge expansion of
domestic production.

cause event 5
predicate: DISAPPROVE
time: 2022-03-03
subject: TRADER
object: RUSSIA
headline: Energy markets in turmoil as European gas climbs 60%
```

Listing 3: A prompt example used for GDELT.

We show one example of the prompt structures used for ICEWS dataset as below. The effect event consists of an event predicate, an event time, a subject name and an object name while the cause events consists of an event sub predicate additionally.

```
I want you to do the reasoning over social events. I given you an effect event
and you give me four or five cause events. An effect event is an event that
happens. A cause event is an event that is believed to be one of the causes that
trigger an effect event to happen. Each event consists of an event text, an event
 predicate, an event time, subject name and object name of describing the event.

The predicates of the effect and cause events are restricted to 20 options, with
names (in capital) and the descriptions below.
1. MAKE STATEMENT: decline comment, make comments.
2. APPEAL: appeal for material, economic, military, humanitarian aid or
cooperation.
```

3. EXPRESS INTENT TO COOPERATE: Express intent to engage in material, diplomatic,
 military aid.
4. CONSULT: make a visit, host a visit.
5. ENGAGE IN DIPLOMATIC COOPERATION: praise or endorse, defend verbally.
6. ENGAGE IN MATERIAL COOPERATION: cooperate economically, militarily, judicially.

7. PROVIDE AID: provide economic, military, humanitarian aid.
8. YIELD: ease admin or political sanctions or military blockade, return and
release.
9. INVESTIGATE: investigate crime, corruption, human rights abuses, military
actions.
10. DEMAND: demand any type of cooperation, aid, reforms, rights, easing of
sanctions.
11. DISAPPROVE: criticize or denounce, accuse of crime, human rights abuses,
complain officially and make lawsuit against.
12. REJECT: reject material, economic, military, judicial operations, requests or
 plans.
13. THREATEN: threaten to reduce aid, to boycott, to reduce or break relations,
to impose sanctions.
14. PROTEST: civilian demonstrations.
15. EXHIBIT MILITARY POSTURE.
16. REDUCE RELATIONS: reduce or break any relations.
17. COERCE: seize or damage properties, impose administrative sanctions or
restrictions.
18. ASSAULT: use of unconventional forms of violence.
19. FIGHT: uses of conventional force.
20. ENGAGE IN UNCONVENTIONAL MASS VIOLENCE.

Now I give you 10 examples of event reasoning. In each example, the first event
is the effect event, the next three to five events are cause events that happen
earlier.

## Example 1

effect
predicate: DISAPPROVE
time: 2022-10-28
subject: CHINA
object: JAPAN

reasoning:
--------------------------
cause event 1
predicate: ENGAGE IN DIPLOMATIC COOPERATION
time: 2022-10-18
subject: Japan
object: USA
sub predicate: Rally support on behalf of.

cause event 2
predicate: ENGAGE IN MATERIAL COOPERATION
time: 2022-10-16
subject: SOUTH KOREA
object: UNITED NATIONS
sub predicate: Cooperate militarily.

cause event 3
predicate: ENGAGE IN DIPLOMATIC COOPERATION
time: 2022-10-13
subject: SOUTH KOREA
object: JAPAN
sub predicate: Praise or endorse.

cause event 4

```
predicate: MAKE STATEMENT
time: 2022-10-10
subject: CHINA
object: JAPAN
sub predicate: Make pessimistic comment.
```

Listing 4: A prompt example used for ICEWS.

The following is a prompt exemplar used for the Amazon Review dataset. Each effect event consists of the product category and event time while each cause event also includes the content of the product review.

```
I want you to do the reasoning over the events that are extracted from online-
shopping review data. I given you an effect event and you give me two to four
cause events. An effect event is an event that happens. A cause event is an event
 that is believed to be one of the causes that trigger an effect event to happen.
 Each event corresponds to an review submitted by the customer, which consists of
 an product category(event type), a product title, an event time, summary text
and review text from the user that describes the feedback of the shopping event.

The product categories are restricted to the following set:
1. Women Shoes,
2. Men Shoes,
3. Men Clothing,
4. Women Clothing,
5. Novelty & More,
6. Men Uniforms, Work & Safety,
7. Women Jewelry,
8. Costumes & Accessories,
9. Men Accessories,
10. Luggage & Travel Gear,
11. Men Watches,
12. Women Accessories,
13. Children Shoes,
14. Children Clothing
15. Shoe, Jewelry & Watch Accessories,
16. Women Watches,
17. Women Uniforms, Work & Safety,
18. Men Surf, Skate & Street,
19. Women Handbags & Wallets
20. Men Jewelry
21. Children Accessories
22. Women Maternity
23. Women General
24. Others

# Example 1

effect event
product category: Luggage & Travel Gear
event time: 2013-10-19

---------------------------------
reasoning:
cause event 1
product category: Novelty & More
product title: Sports Katz Peace Out Socks
event time: 2013-09-24
summary text: Peace Out Socks
review text: We ordered these for soccer for my daughter and they worked out well.
 They are very cute and have held up. They are a quite thick, which is why I only
 gave them 4 stars. They will work better for fall season; I would not recommend
them for summer soccer, if soccer is your intended use.

cause event 2
```

```
product category: Luggage & Travel Gear
product title: Wildkin Pink Retro Kickstart Messenger Bag
event time: 2013-09-09
summary text: Wildkin Kickstart Messenger Bag
review text: My daughter, going to 4th grade, wanted a messenger type bag this
year for school. I searched and searched on Amazon for the perfect bag, but all
of the adult bags seemed huge (like they would definately look funny with her
carrying them). Thank goodness I found this bag. It is the perfect size for her,
and her favorite color!.

cause event 3
product category: Children Clothing
product title: Levi's Girls' 715 Thick Stitch Bootcut Jean
event time: 2013-09-01
summary text: Thick Stitch Bootcut Jeans
review text: I love Levis and this style is sooo cute! My daughter even likes
them! The price was amazing in these, as we found similar Levis at Dillard's for
36. The waist of these Levi in particular run small compared to other Levis of
the same size. During the Levis sale on Amazon, we purchased these, the Sweetie
Skimmers, and the Scooter. All of the others ran almost half an inch larger in
the waist. I worry the larger size would be huge on her, but I think I will try
the larger size, as they must be comfortable on her or she will not wear them.

cause event 4
product category: Children Shoes
product title: New Balance KJ750 Running Running Shoe (Lttle Kid/Big Kid)
event time: 2013-02-02
summary text: Cute and durable!
review text: I ordered these for my 8 year old daughter. She usually destroys
shoes within months, but these have held up!
```

Listing 5: A prompt example used for the Amazon data.

For the full set of prompts used in the paper, please see `scripts/gdelt/gdelt_prompts` and `scripts/amazon/amazon_prompts` in our code.

### E.5 Does GPT remember Amazon Review data?

We have a conversion with GPT-3.5-turbo to verify that it does not remember the exact information of Amazon Review data.

```
input: We are using Amazon Review Data, published by Jianmo Ni, et al., 2018,
which is firstly used in the paper "Justifying recommendations using distantly-
labeled reviews and fined-grained aspects", EMNLP 2019.

The dataset includes millions of reviews (ratings, text, helpfulness votes),
product metadata (descriptions, category information, price, brand, and image
features) in the range May 1996 - Oct 2018. Do you remember any of the review
information if I give you a review id?

GPT-3.5 output: As an AI language model, I am not capable of remembering any
review information unless I have been programmed to store and retrieve specific
pieces of information. However, I can search for and retrieve review information
from the dataset if given a review ID.

input: can you recall any information with reviewer ID A28GK1G2KDXHRP on date 04
17, 2014.

GPT-3.5 output: I apologize, but the Amazon Review dataset does not include
information about the specific date when a review was posted. It only includes
the year and month when the review was posted. Therefore, I cannot provide you
with any information about reviews posted by reviewer ID A28GK1G2KDXHRP on April
17, 2014.
```

```
input: Reviewer A0148968UM59JS3Y8D1M bought Hanes Men's Classic Crew on 01 18,
2014, what review did he wrote?

GPT-3.5 output: I apologize, but as an AI language model, I do not have access to
 real-time or updated information about specific Amazon reviews or purchases made
 by individual reviewers. Additionally, the Amazon Review dataset does not
include information about the specific content of a specific review. It only
includes metadata such as the reviewer ID, product ID, and overall rating.
Therefore, I cannot provide you with the review written by the reviewer
A0148968UM59JS3Y8D1M for the Hanes Men's Classic Crew on January 18, 2014 or any
other date.

input: Reviewer A0148968UM59JS3Y8D1M bought Hanes Men's Classic Crew on 01 18,
2014, his review starts with 'they seem very well made..', can you recall the
complete review he wrote?

GPT-3.5 output: I apologize, but as an AI language model, I do not have access to
 real-time or updated information about specific Amazon reviews or purchases made
 by individual reviewers. Additionally, the Amazon Review dataset does not
include information about the specific content of a specific review. It only
includes metadata such as the reviewer ID, product ID, and overall rating.
Therefore, I cannot provide you with the complete review written by the reviewer
A0148968UM59JS3Y8D1M for the Hanes Men's Classic Crew on January 18, 2014, or any
 other date.

input: on which date did reviewer A0148968UM59JS3Y8D1M buy Hanes Men's 7-Pack
ComfortSoft Briefs?

GPT-3.5 output: Sorry, as an AI language model, I do not have access to personal
information or purchase history of individuals.
```

Listing 6: A conversation with GPT-3.5-turbo to test its memorization about Amazon Review data.

### E.6 Training and Inference Details

In this section, we present the details about training the models and doing inference with them.

**LLMs.**  We use GPT-3-davinci and GPT-3.5-turbo. We set temperature to be 0 to ensure that our results are reproducible.

**Training Base Models.**  For each model, we did grid search and chose the hyperparameters based on their performance on the dev set; see Table 2 for the values of the hyperparameters. For KE, DyRep, ANHP, the main hyperparameters to tune are the entity embedding size $D_1$, predicate embedding size $D_2$, RNN hidden size $D_3$ used in the network and the number of layers $L$ of the attention structure (DyRep and ANHP). In practice, the optimal $D$ for a model was usually $4, 8, 16, 32$; the optimal $L$ was usually $1, 2, 3, 4$. In the experiment, to train the parameters of the base model, we performed early stopping based on log-likelihood on the held-out dev set.

**Retrieval.**  For each event, we use few-shot prompting (see Appendix E.4 for examples) to generate a set of cause events. For each LLM-generated cause, we retrieve the most similar $D$ actual events from the history, following the procedure in section 3.2.

- For GDELT, we set $D = 2$ and the average number of retrieved events is $10$.

- For ICEWS, we set $D = 2$ and the average number of retrieved events is $10$.

- For Amazon Review, we set $D = 4$ and the average number of retrieved events is $10$.

**Training Ranking Model.**  The energy function used in our ranking model is the same as what's proposed in Xue et al. (2022), which consists a continuous-time Transformer and an MLP. The hyperparameters are tuned within a range of values that make the score function to have a similar size as the base ANHP model. Training this ranking model needs negative samples. On GDELT and ICEWS, we use 5 negative samples forpredicate prediction and 20 for object and predicate-object prediction. On Amazon Review, we use 5 for both type and time prediction.

| MODEL | DESCRIPTION | VALUE USED | | |
|---|---|---|---|---|
| | | GDELT | ICEWS | AMAZON |
| KNOW-EVOLVE | ENTITY EMBEDDING SIZE | 16 | 16 | NA |
| | PREDICATE EMBEDDING SIZE | 4 | 4 | NA |
| | RNN HIDDEN SIZE | 16 | 16 | NA |
| DYREP | ENTITY EMBEDDING SIZE | 16 | 16 | NA |
| | PREDICATE EMBEDDING SIZE | 4 | 4 | NA |
| | RNN HIDDEN SIZE | 16 | 16 | NA |
| | ATTENTION LAYERS NUMBER | 1 | 1 | NA |
| NHP | RNN HIDDEN SIZE | NA | NA | 36 |
| | ENTITY EMBEDDING SIZE | NA | NA | 16 |
| | PREDICATE EMBEDDING SIZE | NA | NA | 8 |
| ANHP | ENTITY EMBEDDING SIZE | 16 | 16 | 32 |
| | PREDICATE EMBEDDING SIZE | 4 | 4 | 32 |
| | HEADS NUMBER | 2 | 1 | 1 |
| | ATTENTION LAYERS NUMBER | 1 | 1 | 2 |
| RANKING MODEL | TEMPORAL EMBEDDING SIZE | 40 | 40 | 24 |
| | HIDDEN SIZE | 60 | 60 | 56 |
| | HEADS NUMBER | 4 | 3 | 16 |
| | ATTENTION LAYERS NUMBER | 3 | 3 | 3 |

Table 2: Values of hyperparameters used for models trained on the three datasets.

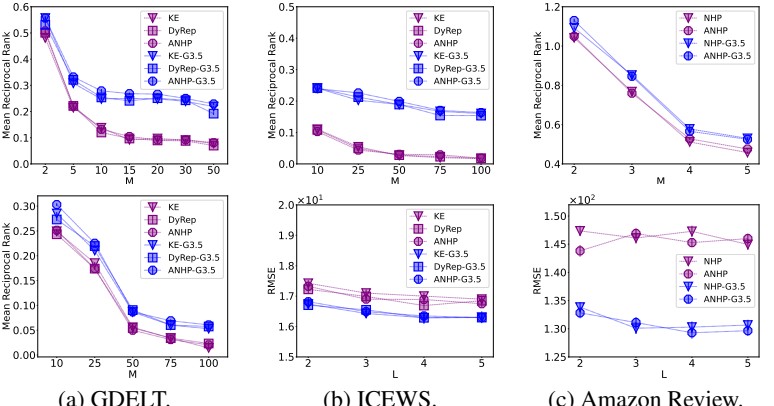

(a) GDELT.  (b) ICEWS.  (c) Amazon Review.

Figure 9: Prediction performance measured by MRR of different methods on each dataset. The configuration of this figure is the same as that of Figure 2: on GDELT, we show object (upper) and predicate-object (lower) prediction; on ICEWS, we show object (upper) and time (lower) prediction; on Amazon Review, we show type (upper) and time (lower) prediction.

**Computation Cost.** All the experiments were conducted on a server with 256G RAM, a 64 logical cores CPU (Intel(R) Xeon(R) Platinum 8163 CPU @ 2.50GHz) and one NVIDIA A100 GPU for acceleration. The training batch size is 8. For GDELT and ICEWS, the wall-clock training time is: 4.9ms per sequence for KE; 6.7ms per sequence for DyRep; 28.7ms per sequence for ANHP. For Amazon Review, the wall-clock time is: 3.6ms per sequence for NHP; 5.2ms per sequence for ANHP. For each of the datasets, the wall-clock time of training the ranking model is 9ms per training sample.

# F   Additional Results and Analysis

In this section, we present additional results and analysis that complement section 4.

### F.1   Main MRR results.

Our main results in section 4 are the MR results. In this section, we show the corresponding MRR results, which are known to be more robust to bad predictions. The MRR results are in Figure 9.

### F.2   Analysis About Precision and Recall

Figure 10 shows the mean average precision (MAP) and mean average recall (MAR) of each base model for a range of $M$. These metrics complement MR and MRR for evaluating the base models: MAP measures the fraction of the predictions that are correct; MAR measures the fraction of the

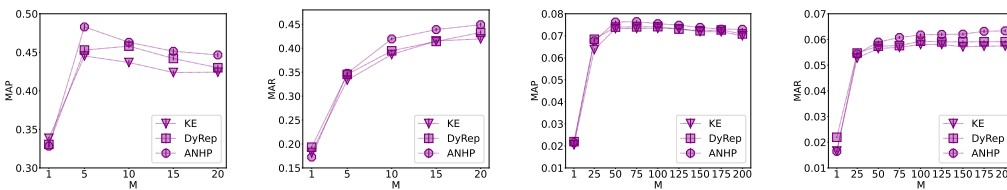

(a) MAP on predicate prediction.  (b) MAR on predicate prediction.  (c) MAP on object prediction.  (d) MAR on object prediction.

Figure 10: MAP@M and MAR@M on GDELT dataset.

actual events that are covered by the predictions; higher is better. Their precise definitions (and formulas) for our setting can be found in Appendix E.2.

As we can see in Figure 10, when $M$ increases, MAR monotonically increases but the change of MAP may not be monotonic (which is consistent with how MAP and MAR behave in other retrieval tasks). For predicate prediction, we found that $M = 5$ yields a good balance between precision and recall as well as a low financial cost (i.e., GPT API calls). For object prediction, $M = 50$ seems to give the best precision-recall balance but we chose $M = 20$ due to the GPT API budget. For object prediction, both MAP and MAR are lower than 0.1 (but still significantly better than a random guess of 0.0015), indicating that this data is very difficult to model.

### F.3 Analysis About Generalization

The LLM in LAMP is instructed by a few demonstrations. On GDELT, these demonstrations cover

- 17 out of 20 (85%) predicates;
- 30 out of 2279 (1.31%) subjects;
- 28 out of 2279 (1.22%) objects.

However, the LLM is able generalize beyond the demonstrations, suggesting cause events that involve novel predicates and entities. Precisely, the LLM-generated cause events involve

- 4699 distinct predicates, which cover all the 20 ground-truth predicates;
- 60552 distinct subjects, which cover 30 (of 30) subjects mentioned in demonstrations, as well as 787 not-mentioned ground-truth subjects.
- 65554 distinct objects, which cover 28 (of 28) objects mentioned in demonstrations, as well as 788 not-mentioned ground-truth objects.

The retrieved cause events cover:

- all the 20 ground-truth predicates.
- 30 (of 30) subjects in demonstrations, as well as 186 not-mentioned ground-truth subjects.
- 28 (of 28) objects in demonstrations, as well as 216 not-mentioned ground-truth objects.

Note that retrievals involve fewer distinct subjects and objects than the LLM-generated causes. It is because the LLM generation is often diverse and creative, proposing many novel subject-predicate-object combinations that involve ground-truth entities but haven't actually happened in real history. Such causes will eventually be grounded to actual previous events that exhibit less diversity and creativity, thus ending up with fewer distinct subjects and objects in the retrievals. Benefiting from the strong generalization capability of LLMs, our LAMP framework has a significant potential for broad applications.

### F.4 Additional Results About LLMs

Now we present our additional results of comparing different LLMs. They include the MR results of comparing GPT-3 and GPT-3.5 on GDELT (Figures 11 and 12), MR results of comparing GPT-3 and GPT-3.5 on Amazon Review (Figure 13), and MRR results corresponding to Figure 3 (Figure 14).

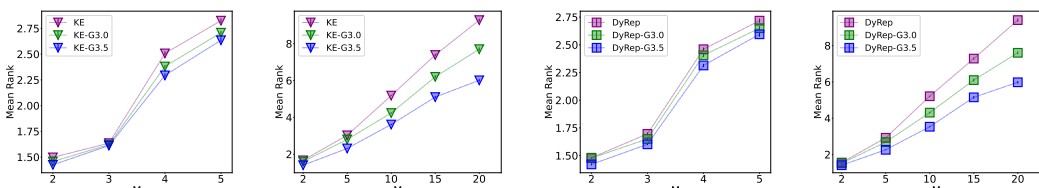

Figure 11: Different LLMs on predicate (left) and object (right) prediction on GDELT dataset, with KE.

Figure 12: Different LLMs on predicate (left) and object (right) prediction on GDELT dataset, with DyRep.

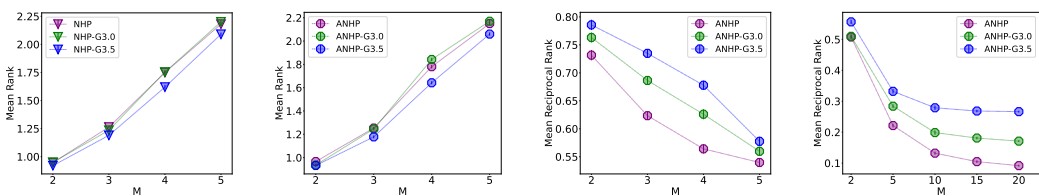

Figure 13: Different LLMs on type prediction on Amazon Review.

Figure 14: MRR results of different LLMs on predicate (left) and object (right) prediction on GDELT.

## F.5 Additional Results About Prompt Design

Now we present additional results about the prompt design analysis in section 4. Figure 15 and Figure 16 show the MRR results corresponding to Figure 5 and Figure 6, respectively.

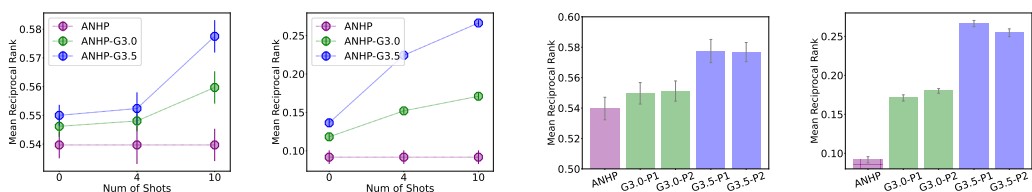

Figure 15: MRR results corresponding to Figure 5: predicate (left) and object (right) prediction on GDELT.

Figure 16: MRR results corresponding to Figure 6: predicate (left) and object (right) prediction on GDELT.

## F.6 Additional Results About Retrievals

Now we present our additional analysis results about retrievals. Figure 17 presents the MRR results with different numbers of retrieved evidence events on GDELT.

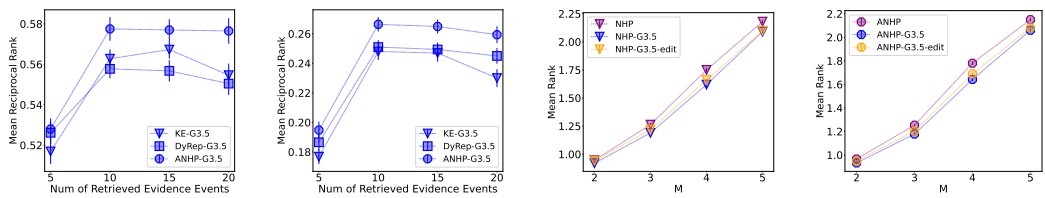

Figure 17: MRR results on GDELT corresponding to Figure 4: predicate (left) and object (right) prediction.

Figure 18: MR results of type prediction with different similarity metrics on Amazon Review.

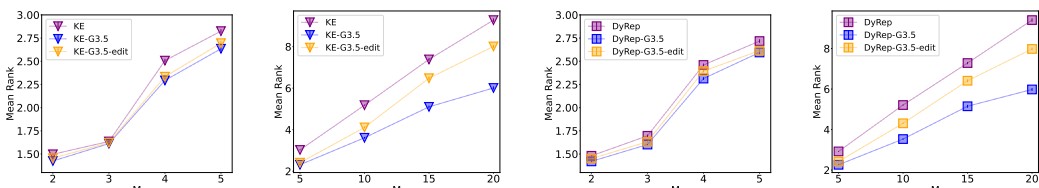

Figure 19: About similarity metric: predicate (left) and object (right) prediction on GDELT, with KE.

Figure 20: About similarity metric: predicate (left) and object (right) prediction on GDELT, with DyRep.