# OpenReview forum: "Language Models Can Improve Event Prediction by Few-Shot Abductive Reasoning"
_NeurIPS.cc/2023/Conference — NeurIPS 2023 poster_

### Official Review · Reviewer_mHgh · 2023-07-03

**Soundness:** 2 fair
**Presentation:** 2 fair
**Contribution:** 2 fair
**Rating:** 6
**Confidence:** 4

**Summary:**

This paper studies using large language models (LLMs) to assist event prediction. The authors proposed a pipeline of procedures, where 1) a traditional event sequence model is first applied to predict a set of possible events along with their time stamps, 2) then for each event candidate, an LLM is used to reason about their cause events, based on which a set of relevant events are retrieved from the history, and 3) finally the retrieved relevant events are used by a reranking model to score each event candidate. This framework is tested on two datasets, GDELT and Amazon Review, and the results showed that the LLM-based reranking approach outperforms the naive event sequence model.

**Strengths:**

1. This work studies leveraging LLMs for temporal modeling. This is an important topic as it can be used to support many applications (e.g., political event forecasting). Using LLMs for reranking also seems to be novel.
2. The authors have conducted experiments on two datasets, both showing an advantage of the LLM-based reranking.
3. This paper also comes with analyses of the proposed framework, such as how tuning the size of event candidates (M) can affect the overall performance and whether the LLM is sensitive to different variants.

**Weaknesses:**

1. The paper needs significant justifications for the experiment design.
- While the authors formulated the event prediction task into two subtasks, the majority of the experiments have focused on only one of them (i.e., event type prediction), whereas the discussion on the time stamp prediction is rather brief. If the authors regard the time prediction task as an important subtask, then they may want to consider using a dataset that can facilitate more careful discussions other than GDELT.
- The event prediction on GDELT is not end-to-end. Instead, the current experiments always assume one or two (ground-truth) components from each <subject, predicate, object> triplet and aim to predict the remaining. This could show a discrepancy between the evaluation and the realistic application.
- More clarifications about each dataset are needed. For example, it is unclear how many events are provided as history in each prediction.

2. Results in Figure 2 and several other figures showed worse performance as M increases. This could say that as the LLM-based reranker is exposed to more candidates, it ranks the truth even worse. Looking into the description of Mean Rank in lines 250-256, it can be imagined that as M increases, there will be more test cases (whose ground truth is included in top-M) get counted into "N". In other words, when M has different values, the Ns have been different, so the results of the same approach across different Ms are not really comparable. However, this is not clarified in the paper and the current plots read confusing.

3. The proposed framework heavily relies on the event sequence model. Based on Figure 3, it seems that the poor performance of the event sequence model is the real bottleneck (<0.1 MAR in Fig 3(b)). In other words, while introducing LLMs does lead to improvement in event reranking (under the current experimental setting), it is not solving the real bottleneck. Therefore, I'm not very convinced by the contribution of this framework. Instead of reranking, a more worth-studying problem is how to use LLMs to more accurately predict future events in the first place.

4. More explorations on the use of LLMs in reranking can provide more insight to the community. It is noticed that, while the contribution of this paper is about using LLMs for event reranking, none of the baselines or variants are following the same reranking idea. Is there a better way to use LLMs for event reranking? For example, can one use SentenceBERT or BM25 to retrieve relevant events from the history, and then directly use LLMs to score the candidate event?

5. Some of the writing can be more self-contained. For example, the paragraph in line 51 introduced several math formulations but there is not a single reference, and the intuition of these formulas is not explained clearly.


**Questions:**

Please try to address my comments in Weaknesses.

**Limitations:**

The authors discussed the limitations and potential negative impact in Appendix.

---

> ### Author Rebuttal · Authors · 2023-08-10
>
> Thank you for your feedback!
>
> > consider using a dataset that can facilitate more careful discussions other than GDELT.
>
> We added new results on a new dataset, ICEWS. It is similar to GDELT but less dense in time so it is meaningful to predict time on this new dataset. Please see [New Dataset] for details.
>
> In the final version, we will include a full set of experiments on ICEWS (all baselines, all analysis) just as we did for GDELT.
>
> > assume one or two (ground-truth) components from each <subject, predicate, object> triplet and aim to predict the remaining. This could show a discrepancy between the evaluation and the realistic application.
>
> Our evaluation setup matches real-world applications. Please see [About Experiment Design].
>
> > More clarifications about each dataset are needed. For example, it is unclear how many events are provided as history in each prediction.
>
> We will add this detail to the final version: "history" includes all the previous events in the data.
>
> > when M has different values, the Ns have been different, so the results of the same approach across different Ms are not really comparable. However, this is not clarified in the paper and the current plots read confusing.
>
> Sharp catch! Yes, our current text in Line-264 to 266 is misleading. We didn't mean to compare the same method across M; instead, we meant to compare different methods for the same M. We apologize and will definitely correct our presentation.
>
> > proposed framework heavily relies on the event sequence model… it is not solving the real bottleneck… a more worth-studying problem is how to use LLMs…
>
> Which's real bottleneck seems to be a subjective call? Please see [Contribution, Significance, and Impact] for our justification of the problem setting and our technical approach.
>
> > none of the baselines or variants are following the same reranking idea. Is there a better way to use LLMs for event reranking?
> > can one use SentenceBERT or BM25 to retrieve relevant events from the history, and then directly use LLMs to score the candidate event?
>
> In [New Baselines], we added several new baseline methods that follow the same reranking idea of our framework.
> One of them retrieves with sBERT and reranks with our method, but it works poorly. That is because sBERT tends to retrieve events that are similar to the proposal (and BM25 will as well), but the real causes may look very different. Fortunately, LLMs may generate real causes that look very different.
>
> We suspect that using LLMs as scoring function will only make the performance worse because
> - LLMs do not seem to be good at numbers as they are good at words;
> - LLMs are pretrained but our proposed reranking module has been trained on the task (so it is more specialized).
>
> > Some of the writing can be more self-contained. For example, the paragraph in line 51 introduced several math formulations but there is not a single reference, and the intuition of these formulas is not explained clearly.
>
> We are sorry that we made this background paragraph too compact due to the page limit. In the final version, we will appropriately label these formulas, refer to them when needed, and clarify the intuition.

---

> > ### Comment · Reviewer_mHgh · 2023-08-13
> >
> > Thank you for your response (to me and other reviewers)! It well addressed my concerns. The newly added experiments are helpful. I'd like to see them along with the justification of the task setting and your contribution in the next draft. I've raised my score.

---

> > > ### Author Response · Authors · 2023-08-14
> > > **Thank you for your support!**
> > >
> > > Thank you very much for your support!
> > >
> > > We are thrilled to hear that our response has addressed your concerns. And we are definitely committed to improving our presentation for the next version.
> > >
> > > Later in Discussion phase, we will post our plans for improving the paper. (We will hold it until then in case that any reviewer requests anything new.)

---

### Official Review · Reviewer_pKDi · 2023-07-05

**Soundness:** 3 good
**Presentation:** 3 good
**Contribution:** 3 good
**Rating:** 6
**Confidence:** 4

**Summary:**

This paper studies the problem of predicting future world events based on the past and proposes an approach that combines the existing event sequence models with the powerful (abductive) reasoning ability of large language models (LLMs) that results in better performance both for predicting the actual event as well as predicting the time of the event.

**Strengths:**

* Predicting future world events based on the past is an important problem.
* The idea of leveraging LLMs for future event prediction is interesting and can spark further research in this direction.
* Experimental results are strong, albeit on a small set of datasets.

**Weaknesses:**

* The LLM prompts presented on Page 4 are a bit worrisome as they hint at a possible leakage: The time of the first demonstrating example is 2022-03-08 and so is the time of the queried effect. Could you please clarify how you selected the fewshots, and whether there might be any leakages?
* I’m a bit confused about the metrics. From what I understand, the performance of the base Event Sequence model is computed on all the data, but the performance of the LLM-augmented models is only computed on the examples where the correct answer was among the top answers of the base model. Is this correct or did I misunderstand something?
* Some of the presented results on the two datasets are strong, but given the simplicity of the proposed solution, it would have been more convincing if results on more datasets were reported, especially because the mean rank results on the Amazon dataset is not that glaring. (To be clear, I like the simplicity of the approach)
* Given that the LLM module is the main novelty of the work, I would expect to see some experiments where the LLM is replaced with 1) a random selection module that selects 10 pieces of evidence at random, 2) a heuristic-based selection module that, e.g., selects the most recent 10 pieces of evidence with the same subject.
* Some qualitative examples are needed to show how the LLM module actually helps make better predictions.
* [Minor] The font in Figure 1 is a bit small when printed. Maybe make the size of box 3 smaller (rotate the text if needed) so that the other parts become slightly bigger.

**Questions:**

* The abductive reasoning module is quite similar to the backward reasoning works such as [1, 2, 3]. Could the authors say a few words about whether it is possible to bring more of backward reasoning into their framework? As an example, can we predict multiple hops of causes instead of one (e.g., for an event E, predicting a cause C and then predicting a cause C’ for C, …)?
* Any idea how all three modules can be trained end-to-end instead of training separate modules?
* Are the Event Sequence Model and Reranking Model both trained on the same subset of the data?
* [Minor] Why not use Mean Reciprocal Rank instead of Mean Rank? The former has been advocated in many previous works.
* [Minor] LLMs can hallucinate causes and it seems like all causes will be mapped to an event. I wonder whether it helps if the authors ignore the causes whose similarity to existing events is below some threshold.
* [Minor] In line 150, how is time represented so that the model can know about the distance between the events?

[1] LAMBADA: Backward Chaining for Automated Reasoning in Natural Language

[2] Neural story planning

[3] Language Models with Rationality (I understand that this paper was published too close to the NeurIPS deadline)

**Limitations:**

* It is typical in real-world problems to have both temporal and non-temporal facts/events. Can the approach be extended to this case?
* It is typical in real-world problems to have temporal events that have a timestamp and temporal events that have a time interval. Can the approach be extended to work with intervals?
* Can the model predict that no event will happen?

---

> ### Author Rebuttal · Authors · 2023-08-10
>
> Thank you for your constructive feedback and being supportive! We added new results for you; please see [New Baselines] and [New Dataset]. Now we'll answer your remaining questions.
>
> > The time of the first example is 2022-03-08 and so is the time of the queried effect… clarify how you selected the fewshots, … any leakages?
>
> We randomly sampled training events and manually found evidence/cause events for each of them.
>
> Raw data has HH-MM-SS information so we know the temporal order of same-day events. For presentation simplicity, we omit HH-MM-SS in examples. We'll clarify it (or use a better example) in final version.
>
> We have taken efforts to ensure that there is no leakage in data. For GDELT and ICEWS, we use the data that is surely not in GPT training data. App C.5 in paper explains why GPTs do not memorize Amazon Review data. For each dataset, we split train/dev/test based on time-stamps of events (see Line 214).
>
> > I’m a bit confused about the metrics…
>
> All methods are evaluated where the correct answers were among the top answers of the base model. That is why we feel necessary to show MAP and MAR of base models (see Figure-3): MAP and MAR measures precision and coverage of each base model; mean rank and RMSE measure how much better our framework could do given that precision and coverage of base model.
>
> > it would have been more convincing if results on more datasets were reported
>
> Sure! Please see our new results in [New Dataset], and we hope you like them!
>
> > 1) selects 10 pieces of evidence at random, 2) selects the most recent 10 pieces of evidence with the same subject…
>
> We added these new baselines (and more), and they all underperform our framework; see [New Baselines].
>
> > font in Figure 1 is a bit small… make the size of box 3 smaller (rotate the text if needed)...
>
> Great suggestion! We'll do it in the final version.
>
> > say a few words about whether it is possible to bring more of backward reasoning into their framework?
> > predict multiple hops of causes instead of one (e.g., for an event E, predicting a cause C and then predicting a cause C’ for C, …)?
>
> We drafted a long and in-depth discussion for these questions, but we couldn't show it in this message due to 6000-char limit. Here we only give TLDR, but we can post full discussion via an Official Comment in Discussion phase, if you are interested. We will add full discussion in final version.
>
> Abductive reasoning is a type of reasoning ability, in parallel with deductive reasoning and inductive reasoning. Backward chaining is a general method for reasoning, in parallel with forward chaining and resolution.
>
> In this work, we leverage abductive reasoning abilities of LLMs, and our proposed method resembles backward chaining (although it only does 1-step reasoning).
>
> When data is complete (i.e., full observed, no missing events), 1-step seems to be good enough: both C' and C are in history, and retrieving C seems more proper since C is more recent.
>
> When data is incomplete, multi-step reasoning seems to be necessary: maybe C is not observed in history but C' is, so we have to do a 2nd step reasoning and retrieve C'. This is an extension of our current method.
>
> > Any idea how all three modules can be trained end-to-end
>
> Yes. Conceptually, one can do:
> - use retrieved causes to regularize (e.g., attention mechanism of) event sequence models
> - prompt LLM so that it generates cause events that tend to have high reranking scores
> These can be future directions and we will discuss them in the final version.
>
> > … both trained on the same subset of the data?
>
> The use same set of training data. But reranking uses negative samples drawn from event model. They are both early-stopped based on dev results.
>
> > Why not use Mean Reciprocal Rank
>
> We thought Mean Rank is the norm of the IR community. We can add Mean Reciprocal Rank in the final version.
>
> > I wonder whether it helps if the authors ignore the causes whose similarity to existing events is below some threshold.
>
> We tried this idea and showed new results in [New Baselines]. There is a small difference (thresholding worse), but a heavier tuning of threshold may close the performance gap.
>
> > In line 150, how is time represented… model can know about the distance between the events?
>
> In the continuous-time Transformer of Yang et al. 2022, time is represented via a temporal embedding mechanism (a generalization of positional embedding) so high-layer representations are aware of the temporal distances between events.
>
> > It is typical in real-world problems to have both temporal and non-temporal facts/events. Can the approach be extended to this case?
>
> Yes, we believe that the general propose-LLM-retrieve-rerank pipeline can apply to other settings, although they may need non-trivial technical extensions. Here is how it may apply to language-based logical reasoning: given a theory of multiple statements, a base model can propose possible conclusions; LLM proposes possible explanations for each proposed conclusion; retriever finds out most relevant statements from the theory; reranking module examines if the retrieved statements can actually prove the proposed conclusion.
>
> We will add this discussion in our final version.
>
> > Can the approach be extended to work with intervals?
>
> If the base event sequence model is capable of handling intervals, then it seems straightforward for our framework to handle intervals as well (through the base model).
>
> > Can the model predict that no event will happen?
>
> That is the duty of base event sequence model. If the most recent event is at time t0, and the base model predicts the next event at time t1, then any time in (t0, t1) is regarded to be "no event happens here". The base models we use in this paper are all probabilistic point processes, so "no event will happen ever from now" has zero probability under the base model.
>
> > qualitative examples needed
>
> We'll add them in final version. See [Qualitative Results].

---

> > ### Comment · Reviewer_pKDi · 2023-08-10
> > **Reviewer Response**
> >
> > Thanks for the detailed response, and for providing the extra analysis and results.
> >
> > * **Regarding leakage**: Thanks for clarifying. My main concern with respect to leakage is now resolved. Although I still believe it would be better to considered a larger time gap than a few minutes or hours between train and test sets to ensure no unwanted leakage. Maybe providing some statistics on the time difference quantiles between test set questions and the latest fewshot example could further resolve the issue.
> >
> > * **Regarding baselines**: In the case of the ANHP-rec-10-evt baseline, are you using the 10 most recent events with the same subject, or just the 10 most recent events? If the former, do you have any intuition why it is performing so poorly (even worse than random)?
> >
> > * **Regarding single vs multi-hop**: I found your response on multi-hop mostly being helpful when the data is incomplete quite helpful and insightful. I strongly suggest adding it to the next draft.
> >
> > * **re min rank vs mean reciprocal rank**: The problem with min rank is that one bad prediction can substantially outweigh many good predictions. This is not the case for mean reciprocal rank, though. So I often find comparisons in terms of MRR more meaningful than MR.

---

> > > ### Author Response · Authors · 2023-08-13
> > > **New Results - Leakage, 10-Most-Recent-Same-Subject, MRR**
> > >
> > > We highly appreciate your engagement and prompt reply!
> > > We ran more experiments and hope the new results further address your concerns.
> > >
> > > [More Results for Leakage]
> > >
> > > > consider a larger time gap… statistics on the time difference quantiles between test set questions and the latest fewshot example…
> > >
> > > Definitely. Our few-shot demonstrations cover a wide time range of training data, and there is indeed a big gap between train and test. Precisely, the latest demonstration is on 2022-06-07, and the earliest test event is on 2022-07-15 (and dev events stay in-between). Below is a table of percentiles (in days):
> > >
> > > |    0%    |    1%    |    5%    |    25%    |    50%    |    75%    |    95%    |    99%    |
> > > |-----------|-----------|-----------|------------|--------------|------------|------------|-------------|
> > > |     38    |    40     |     41    |      44     |      52      |     54      |      55     |      55     |
> > >
> > > [Recent Events with Same Subject]
> > >
> > > > 10 most recent events with the same subject, or just the 10 most recent events?
> > > > do you have any intuition why… even worse than random
> > >
> > > ANHP-rec-10-evt only uses the 10 most recent events, and we just experimented with ANHP-rec-sub-10-evt, a new baseline that uses the 10 most recent events with the same subject. Their MRR results are in [MRR Results]. The takeaways are:
> > > - ANHP-rec-sub-10-evt is slightly (but almost negligibly) better than ANHP-rec-10-evt.
> > > - both are worse than random.
> > >
> > > Why does "most recent" work so poorly? It is perhaps because "recent" is a bad inductive bias. On dev data, we analyzed the cause events retrieved based on LLM-generated clues and found that many of them are not "recent": for each dev event, we track the time gap between it and its closest retrieved evidence event; then we compute the percentiles of these time gaps; the table of percentiles (in days) is shown below.
> > >
> > > |   5%   |    25% |    50%    |    75%    |    95%   |
> > > |-----------|-----------|-----------|------------|--------------|
> > > |  0.54  |  2.15   |   3.81     |   24.79   |   38.66   |
> > >
> > > The "same subject" inductive bias is not very helpful for predicate prediction. When predicting predicates, the subject and object are known, and thus the negative samples (given by base model) will all have the same subject. As a result, all the candidates will have the same set of "evidence", making it very difficult for the reranking model to learn to figure out which predicate is correct.
> > >
> > > However, "same subject" inductive bias may be helpful for object prediction and predicate-object joint prediction. Therefore, we have been evaluating the new baselines on these kinds of prediction tasks, and we will post an update as soon as possible.
> > >
> > > [MRR Results]
> > >
> > > > MRR more meaningful than MR.
> > >
> > > Lessons learned. Thanks! We evaluated all methods with MRR as well, and the ranking of methods remains the same under this metric; please see the table below for predicate prediction results on GDELT.
> > >
> > > | M                                            |    2             |      3          |       4         |        5          |
> > > |----------------------------------------|----------------|---------------|---------------|-----------------|
> > > |  ANHP                                    |   0.7316     |    0.6234   |   0.5641    |     0.5397    |
> > > |  ANHP-rnd-10-evt                  |   0.7185     |    0.6013   |    0.5250    |    0.4675     |
> > > |  ANHP-rec-10-evt                  |    0.6078    |   0.4301    |     0.3534   |     0.3106    |
> > > |  ANHP-rec-sub-10-evt           |    0.6105    |   0.4340    |     0.3601   |     0.3121    |
> > > |  ANHP-bert-10-evt                 |   0.7263     |    0.6264   |    0.5629    |     0.5278    |
> > > |  ANHP-text-emb                    |  0.7330      |    0.6233   |   0.5650     |      0.5401    |
> > > |  ANHP-llama                         |   0.7631     |     0.6673   |    0.6004    |     0.5533     |
> > > |  ANHP-G3.5(ours)                 |   0.7775     |    0.6868    |     0.6217  |      0.5775    |
> > >
> > > The takeaways remain the same: our framework is better than all baselines; GPT-3.5 is better than LLAMA2-chat-13B, which is better than others; etc.

---

> > > ### Author Response · Authors · 2023-08-13
> > > **New Discussion - More About Logical Reasoning**
> > >
> > > [More About Logical Reasoning]
> > >
> > > > quite helpful and insightful. I strongly suggest adding it to the next draft.
> > >
> > > Thanks! We are glad that you like it!
> > >
> > > We will surely add it to the paper; as we said, we have already written the full paragraph. We plan to add it as a new section: 3.4 Relations with Formal Logical Reasoning, but may also move it to Related Work (since it will cite many papers, including what you've suggested). We post our current draft here; please let us know if it needs any improvements.
> > >
> > > <-- paragraph about our method vs. logical reasoning
> > >
> > > %%% to keep rebuttal concise, we omit references here
> > >
> > > Our proposed method is deeply connected with the research on formal logical reasoning. In formal logical reasoning, one will be given a theory---i.e., a set of facts and rules written in formal language (e.g., prolog)---and asked to determine the truth value of a goal---i.e., a logical statement which may be a fact. A typical method for this problem is backward chaining: it starts from the goal and works backward to determine the sequence of steps needed to prove the goal, by applying the rules and identifying relevant facts as preconditions. This method has been generalized to solving other AI problems, including reasoning and planning in natural language. In our problem of future event prediction, each proposal given by the base model is a goal, and the full history can be regarded as a theory that only contains facts, i.e., the events that have actually happened. Like backward chaining, our method aims to find out the preconditions (i.e., cause events) for each goal. The key innovation is: our method utilizes the abductive reasoning capability of LLMs---precisely, the ability to reason about explanations for an outcome---to propose possible causes, which it then pattern-matches against to retrieve the actual events from the history. As a result, our method doesn't need any explicit rules, but relies on the built-in knowledge of LLMs. Notably, unlike backward chaining, our method does not perform more than one step of reasoning. It is because we assume that the data is complete: i.e., all events that have happened are observed. In this case, a second step of reasoning is unnecessary since the direct causes are all observable, thus retrievable, and they are more recent in time than any indirect causes. When the data is incomplete, it will be necessary to perform multiple steps of reasoning, since the direct causes may not be observable and one has to find out indirect causes from the history. Handling incomplete data will be a non-trivial extension of our proposed method and we leave it to future work.
> > >
> > > -->

---

> > > ### Author Response · Authors · 2023-08-17
> > > **New Results - 10-Most-Recent-Same-X**
> > >
> > > We finished new experiments and here come the new results!
> > >
> > > First, I need to correct my statement of "'same subject' inductive bias may be helpful for object prediction and predicate-object joint prediction". That is wrong: for object and predicate-object joint prediction, "same subject" will also retrieve the same set of evidence for all proposals, making it hard to learn for the reranking model. Sorry for this mistake.
> > >
> > > The "same X" inductive bias should help more when the retrievals are different across proposals. Such settings include:
> > > - predicate prediction: retrieve 10 most recent events with the same predicate;
> > > - object prediction: retrieve 10 most recent events with the same object.
> > >
> > > Now let's look at the new results in these settings.
> > >
> > > First, we recap the table for predicate prediction on GDELT data, adding a new baseline called ANHP-rec-pre-10-evt (i.e., retrieving 10 most recent events with the same predicate).
> > > As we can see, it outperforms "random retrieval" and other "most recent" designs for most values of M; but it is not as good as stronger baselines such as ANHP-bert-10-evt.
> > >
> > > | M                   | 2       | 3      | 4      | 5      |
> > > |---------------------|---------|--------|--------|--------|
> > > | ANHP                | 0.7316  | 0.6234 | 0.5641 | 0.5397 |
> > > | ANHP-rnd-10-evt     | 0.7185  | 0.6013 | 0.5250 | 0.4675 |
> > > | ANHP-rec-10-evt     | 0.6078  | 0.4301 | 0.3534 | 0.3106 |
> > > | ANHP-rec-sub-10-evt | 0.6105  | 0.4340 | 0.3601 | 0.3121 |
> > > | ANHP-rec-pre-10-evt | 0.6716  | 0.6202 | 0.5340 | 0.5108 |
> > > | ANHP-bert-10-evt    | 0.7263  | 0.6264 | 0.5629 | 0.5278 |
> > > | ANHP-text-emb       | 0.7330  | 0.6233 | 0.5650 | 0.5401 |
> > > | ANHP-llama          | 0.7631  | 0.6673 | 0.6004 | 0.5533 |
> > > | ANHP-G3.5(ours)     | 0.7775  | 0.6868 | 0.6217 | 0.5775 |
> > >
> > > Then, we ran new experiments for object prediction on GDELT. For this setting, we tried ANHP-rec-obj-10-evt (i.e., retrieving 10 most recent events with the same object).
> > > As shown in the table below, ANHP-rec-obj-10-evt is better than "random retrieval" and other "most recent" designs, although it is significantly worse than our ANHP-G3.5. Similar to predicate prediction, the "same subject" design is worse than "random retrieval", although the gap is not as large as for predicate prediction (this task is harder, so performances are all lower, making the gap smaller).
> > >
> > > | M                   | 2      | 5      | 10     | 15     | 20     |
> > > |---------------------|--------|--------|--------|--------|--------|
> > > | ANHP                | 0.5078 | 0.2214 | 0.1321 | 0.1047 | 0.0917 |
> > > | ANHP-rnd-10-evt     | 0.5044 | 0.2096 | 0.1208 | 0.0854 | 0.0665 |
> > > | ANHP-rec-10-evt     | 0.5001 | 0.2009 | 0.1001 | 0.0666 | 0.0501 |
> > > | ANHP-rec-sub-10-evt | 0.5002 | 0.2005 | 0.1013 | 0.0685 | 0.0522 |
> > > | ANHP-rec-obj-10-evt | 0.5117 | 0.2318 | 0.1502 | 0.1307 | 0.1260 |
> > > | ANHP-bert-10-evt    | 0.5016 | 0.2385 | 0.1768 | 0.1651 | 0.1628 |
> > > | ANHP-text-emb       | 0.5101 | 0.2290 | 0.1336 | 0.1041 | 0.0920 |
> > > | ANHP-llama          | 0.5503 | 0.3287 | 0.2676 | 0.2644 | 0.2612 |
> > > | ANHP-G3.5 (ours)    | 0.5568 | 0.3321 | 0.2789 | 0.2684 | 0.2664 |
> > >
> > > We hope that our new results have fully resolved your concerns. If you have any further questions, please let us know and we will do our best to address them.

---

> > > > ### Comment · Reviewer_pKDi · 2023-08-17
> > > > **Reviewer Response**
> > > >
> > > > Thanks for providing the new results and insights. My concern about the leakage is now fully resolved and I have increased my "sound-ness" score to reflect this. Moreover, the new results you provided during the discussion period strengthen your claims and so I increased my overall rating to reflect that.

---

> > > > > ### Author Response · Authors · 2023-08-17
> > > > > **Thank you!**
> > > > >
> > > > > Thank you very much for your support!
> > > > >
> > > > > We are thrilled to know that our responses have positively affected your opinions.
> > > > >
> > > > > We are committed to improving our presentation for the final version.
> > > > >
> > > > > Later in the Discussion phase, we will post our plans for improving the paper. (We will hold it until then in case that any reviewer requests anything new.)

---

### Official Review · Reviewer_Hwvw · 2023-07-05

**Soundness:** 3 good
**Presentation:** 2 fair
**Contribution:** 3 good
**Rating:** 6
**Confidence:** 4

**Summary:**

This submission proposes a large language model (LLM-) based approach to enhancing event prediction methods. Instructed by a few annotated demonstrations, a large language model is used to suggest possible causes for a proposal to be predicted. Then a search module is used to find out previous events that match the suggested causes. Finally, a scoring function is exploited to evaluate whether the retrieved events could actually cause the proposal. The above three steps can be used to enhance an existing event prediction method. Experimental results on two benchmark datasets (Amazon Review and GDELT) demonstrate that the performance improvement is significant by comparing enhanced methods with baseline ones.

**Strengths:**

Summary:
(1) A new idea for using LLMs to enhance event prediction is proposed.
(2) There are some empirical results for justifying the proposed idea.

**Weaknesses:**

Summary:
(1) The effectiveness of the proposed approach, when being applied to other datasets or different LLMs that are trained by other corpora, is doubt.
(2) The detail about how the proposed approach is used to enhance a baseline method for event prediction is clear.
(3) The description of the proposed methodology lacks of rigor, with some important notations undefined.
(4) The proposed approach has some potential limitations not mentioned in the submission (see below in the Limitation part).

Soundness:
The prediction performance of the proposed approach depends on the quality of the suggested causes by LLMs. If the suggested causes differ a lot from any events in the experimental dataset, the proposed approach may not help. Thus, there is a doubt whether the prediction performance improvement gained by the proposed approach still exists for other datasets or different LLMs that are trained by other corpora. In other words, the proposed approach seems not always sound in terms of improving performance for event prediction.

Presentation:
Although the basic idea is presented clearly in the submission, the details are confusing in some important aspects. First of all, it is unclear how to enhance an existing event prediction method by the three steps (see comments in the Summary part). Should the existing event prediction method be treated as the scoring function in the last step? In other words, does an enhanced method only differs from the baseline one in retrieving relevant events to conduct a prediction? Secondly, the template for constructing prompts to an LLM is unclear. There are two templates (Listing 1 and Listing 2 in page 4) presented in the submission, which one is used to construct a prompt? Finally, some descriptions of the proposed approach are unclear. In Lines 89-90, what does the superscript (1), …, (M) mean? How is M defined? In Line 113, what does the superscript (m) mean? In Line 117, what does the superscript (m,M’) mean? How is M’ defined? In Line 170, what does the superscript (m) mean? How is M in Equation (5) defined?

Contribution:
The submission presents a new idea for utilizing LLMs to enhance event prediction. However, the soundness of this idea is questionable (see comments in the Soundness part). It calls for further theoretical analysis and empirical study to confirm the soundness.

**Questions:**

The questions have been raised in the weakness comments for Presentation.

**Limitations:**

The authors have not adequately addressed all limitations in the proposed approach. Some of them may not be addressed by the state-of-the-art techniques, but authors need to explicitly mention them in the submission. Firstly, the problem setting studied in the work restricts all considering event types to be in a predefined finite set. If the template for constructing a prompt to LLMs is something like Listing 1 in Page 4, this predefined set of event types should be rather small, otherwise the constructed prompt could be too long to feed into the LLM. Secondly, the names of event types should be meaningful, since they need to be fed into an LLM for generating their embeddings. Finally, the effectiveness of the proposed approach depends on whether the utilized LLM involves knowledge about the considering event types.

---

> ### Author Rebuttal · Authors · 2023-08-10
>
> Thank you for your constructive feedback. We have added new results for you; please see [New Baselines] and [New Dataset]. Now we'll answer your remaining questions.
>
> > the soundness of this idea is questionable (see comments in the Soundness part). It calls for further theoretical analysis and empirical study to confirm the soundness.
> > effectiveness of the proposed approach, when being applied to other datasets or different LLMs that are trained by other corpora, is in doubt.
> > If the suggested causes differ a lot from any events in the experimental dataset, the proposed approach may not help… the proposed approach seems not always sound in terms of improving performance for event prediction.
>
> We reply to the "soundness" concern in [Scientific Rigor] of our to-all message.
>
> As for "further empirical study", we have managed to add new results for you and hope that you like them! Please see [New Baselines] and [New Dataset]. First, we experimented with open-source LLAMA2-chat-13B, which was very different from OpenAI's GPTs. Second, we evaluated a new ICEWS dataset, on which we tried both time and predicate prediction. At least for the new model and new data, our proposed framework still exhibits superior performance compared to baseline methods.
>
> Moreover, the LLMs used in our work are all trained with general online data, which has a broad coverage and does not specialize in the particular kinds of data/tasks that this paper is about. Therefore, it seems reasonable to expect that the success can generalize to other test datasets.
>
> > The proposed approach has some potential limitations not mentioned in the submission
> > the problem setting restricts all considering event types to be in a predefined finite set
> > the names of event types should be meaningful, since they need to be fed into an LLM
> > effectiveness of the proposed approach depends on whether the utilized LLM involves knowledge about the considering event types
>
> Thanks. We will include these limitations in the final version.
>
> However, we'd like to point out that the "finite" limitation comes from the base model part of our framework, but not our new LLM and ranking mechanism: nearly all the event sequence models assume a finite set of event types, so our framework still has a broad applicability. Generalization to infinite event types could be a very interesting future direction.
>
> > First of all, it is unclear how to enhance an existing event prediction method by the three steps (see comments in the Summary part). Should the existing event prediction method be treated as the scoring function in the last step? In other words, does an enhanced method only differs from the baseline one in retrieving relevant events to conduct a prediction?
>
> The 3-step framework is shown in Figure-1 of paper, and it is qualitatively described in line-33 to 41 in Introduction. In the final version, we will add a high-level technical introduction at the beginning of Section 3 to describe how the 3 steps work together, and clarify where the event sequence model is used:
> - 1st step: an even sequence model is used as a proposer to propose predictions on the future (time, type, or attribute of type). This is where the existing event prediction model is used. This is also why we call an event sequence model "base model": if we do not do step-2 and 3, this framework falls back to an ordinary event sequence model.
> - 2nd step: LLM reads each proposed prediction, and generates possible causes as if it actually happened. Then sBERT-based retriever finds out events that are similar/relevant to the generated causes from the history. See Section-3.1 and 3.2 in paper.
> - 3rd step: a reranking module examines the combos of (proposal, retrieved evidence) and reranks them. See Section-3.3 in paper. This module has to score sequences of time-stamped events, but it doesn't need to propose events. Therefore, its architecture is like an event sequence model on the input side (which may have confused you? we can clarify it in the final version!), but different on the output side. Technically, it is an energy function just like the ranking modules in non-autoregressive sequence models (e.g., HYPRO by Xue et al. 2022).
>
> > the template… is unclear… two templates (Listing 1 and Listing 2 in page 4)... which one is used to construct a prompt?
>
> Listing 1 is the general template that includes instructions and demonstrations. Listing 2 is an example of demonstration. Listing 2 is used in Listing 1. (In Listing 1, we wrote "// Examples are in Listing 2.")
>
> > description of the proposed methodology lacks of rigor, with some important notations undefined.
> > In Lines 89-90, what does the superscript (1), …, (M) mean? … In Line 113, what does the superscript (m) mean? In Line 117, what does the superscript (m,M’) mean? How is M’ defined? In Line 170, what does the superscript (m) mean? How is M in Equation (5) defined?
>
> We apologize for the confusion. We'll clarify the notation details in the final version.
>
> Throughout the paper, we use little letters (e.g., m) as indices and use capitalized letters (e.g., M) as the upper bounds of the indices.
> Specifically, M is the # of proposals and the variable m indexes the m-th proposal.
> Double index (m,m') shows up because each of the M time proposals corresponds to M' different type proposals at that time: i.e., (m,m') indexes the m'-th type proposal at the m-th time proposal.
> Line-50 to 64 in Section 2 explains the relation between time and type prediction in the setting of classical event sequence modeling.

---

> > ### Comment · Reviewer_Hwvw · 2023-08-17
> > **Issue on limited number of involved event types**
> >
> > Thanks for the detailed response, which addressed most of my main concerns.
> > There is a question remained about the number of event types that can be used. Since the names and descriptions of all involved event types should be added to the prompt fed into the LLM, the number of event types is restricted not only by the original dataset but also by the limited size of a prompt to LLM. For example, a model that does not use LLM can deal with 1000 event types but when it integrates with an LLM, it can only construct a prompt to LLM with no more than 100 event types. Thus, how will the proposed approach enhance a basedline model which originally works well for thousands of event types, but the names and descriptions of all these event types cannot be fed into the LLM?

---

> > > ### Author Response · Authors · 2023-08-17
> > > **We handled a hundred millions of event types in our experiments.**
> > >
> > > Thank you for your response!
> > >
> > > We thought your concern was the "finite" set of event types. If the concern is a "small" set of event types, we can reassure you that our approach can indeed scale up to a "large" set of event types. In our GDELT experiments, it already handled **a hundred millions of event types** (2279 subjects x 20 predicates x 2279 objects; see Line-243, Line-244, and Appendix C.1 of paper).
> > >
> > > Maybe what has confused you is our example in Listing-1? In Listing-1, our prompt includes "... predicates are restricted to the 20 options below: 1. MAKE STATEMENT…"
> > >
> > > First, we need to clarify terminologies
> > > - event type: In the field of event sequence modeling, an event type is typically defined as a combination of (subject, predicate, object) such as (biden, make statement, russia). A dataset such as GDELT and ICEWS typically has more than millions of possible event types since it has tens of predicates and thousands of subjects and objects.
> > > - predicate: In both NLP and event sequence modeling, a predicate means information about a subject or an action that subject takes. In the event-centric NLP community, people sometimes use "predicate" and "event type" interchangeably, because they tend to think an "predicate" is a "type of event" that can happen to a subject. In this paper and rebuttal, we follow the convention of event sequence modeling, treating "predicate" only as an attribute of "event type" but not using them interchangeably.
> > >
> > > From our understanding, you seem to think that "predicate" and "event type" are equal. So we feel eager to clarify this terminology.
> > >
> > > Second, our prompt never needs to mention the set of possible subjects or objects, but our approach works remarkably well on object prediction and predicate-object joint prediction; please see Fig-2(a), Fig-3(b), Fig-4(right), Fig-5(right), Fig-6(right), Fig-7(right), Fig-8(right), Fig-9(right) in main paper along with other figures in appendices. The 2279 subjects and 2279 objects (they are the same set) include a large diversity of political entities (orgs, persons, etc) across G20 countries, so it is really challenging to predict them.
> > >
> > > Third, we ran new experiments on GDELT, and found that our method still works very well if we delete this list from the prompt (i.e., no set of predicates is mentioned at all). We show the new predicate prediction results in the table below, where ANHP-G3.5-noprelist is our framework without the content of "predicates are restricted to the 20 options…" As you can see, changes of results are negligible.
> > > | M                   | 2       | 3      | 4      | 5      |
> > > |-|-|-|-|-|
> > > | ANHP-G3.5-noprelist     | 0.7771  | 0.6878 | 0.6187 | 0.5744 |
> > > | ANHP-G3.5 (ours)     | 0.7775  | 0.6868 | 0.6217 | 0.5775 |
> > >
> > > Fourth, we need to highlight: as stated in our response to E9bj, we didn't tune prompts. We designed prompt templates before running any experiments, and haven't changed it since then. We decided to keep the list of "20 options" in the prompt because they are overall short. We will discuss the new results (without this list) in the final version, but we prefer keeping it as the default design of our framework, considering that the full list is really not long.
> > >
> > > There are other reasons why it is not an issue to include the list of predicates in the prompts:
> > > - the set of predicates is usually small. GDELT is one of the largest event databases in the world, and their creators somehow found it sufficient to use only 20 predicates (more precisely, 20 CAMEO codes; ICEWS uses the same CAMEO codes). In event-centric NLP, people work with the "event trigger/type set" (i.e., "predicate set" in our terminology) of ACE 2005 Corpus, which defines 33 "event types" (i.e., "predicates" in our terminology). As you have noted, our method can easily handle <100 predicates in its prompts, which is more than enough for real-world large-scale applications.
> > > - LLMs keep evolving and their context windows will surely become larger. We expect that LLMs will be able to handle >100 predicates in the near future.
> > >
> > > We hope that this long message has resolved your concerns regarding "small set of event types". If you have any further concerns or questions, please let us know and we will be more than willing to discuss them.

---

> > > > ### Comment · Reviewer_Hwvw · 2023-08-18
> > > >
> > > > Thanks for the detailed response again. Yes, in previous comments I referred to predicates in Listing 1 as event types, which introduces some confusion.
> > > > I am happy to know  "new experiments on GDELT, and found that our method still works very well if we delete this list from the prompt (i.e., no set of predicates is mentioned at all). "
> > > > If the list about predicate names and descriptions can be removed from the prompt to LLM, the proposed approach should have wider applications.
> > > > But I still have a question that the predicate names should appear in event causes suggected by the LLM; if the predicate list is removed, can the LLM output predicate names that are not mentioned in the examples (Listing 2) contained in the prompt? If the answer is no, then Listing 2 should contain at least one example for an individual predicate name, right? This way will result in a long prompt as writing all predicate names and descriptions in the prompt. If the answer is yes, then why can an LLM guess reasonable predicate names without seeing them beforehand?

---

> > > > > ### Author Response · Authors · 2023-08-18
> > > > > **LLMs can generate predicates and entities that are not in few-shot examples**
> > > > >
> > > > > Thank you for your quick response and engagement.
> > > > >
> > > > > > Yes, … referred to predicates in Listing 1 as event types, which introduces some confusion.
> > > > > > I am happy to know… still works very well
> > > > > > … proposed approach should have wider applications.
> > > > >
> > > > > We are really glad that we have resolved this concern!
> > > > >
> > > > > > predicate names should appear in event causes suggested by the LLM
> > > > > > can the LLM output predicate names that are not mentioned in the examples (Listing 2)
> > > > >
> > > > > Yes, it can! Let's look at some statistics.
> > > > >
> > > > > First, the few-shot examples do not cover all predicates or entities:
> > > > > - they cover 17 / 20 (= 85%) predicates
> > > > > - their subjects cover 30 / 2279 (= 1.31%) entities
> > > > > - their objects cover 28 / 2279 (= 1.22%) entities
> > > > >
> > > > > Second, LLMs are able to generate predicates that are not in the few-shot examples. Precisely, on GDELT dev set, the LLM-generated cause events involve 4699 distinct predicates, which cover all the 17 predicates in the few-shot examples as well as all the 20 ground-truth predicates. That is, 3 not-mentioned ground-truth predicates are generated.
> > > > >
> > > > > We also checked the subjects and objects that LLM generated. The results are:
> > > > > - LLM generated 60552 distinct subjects, which cover 30 (of 30) subjects mentioned in few-shot examples, as well as 817 (of 2279) ground-truth subjects. 787 not-mentioned subjects are generated.
> > > > > - LLM generated 65554 distinct objects, which cover 28 (of 28) objects mentioned in few-shot examples, as well as 816 (of 2279) ground-truth objects. 788 not-mentioned objects are generated.
> > > > >
> > > > > Third, the retrieved cause events exhibit the following statistics:
> > > > > - they cover 17 (of 17) predicates mentioned in few-shot examples, as well as 20 (of 20) ground-truth predicates. 3 not-mentioned ground-truth predicates are retrieved.
> > > > > - they cover 30 (of 30) subjects mentioned in few-shot examples, as well as 216 (of 2279) ground-truth subjects. 186 not-mentioned subjects are retrieved.
> > > > > - they cover 28 (of 28) objects mentioned in few-shot examples, as well as 244 (of 2279) ground-truth objects. 216 not-mentioned objects are retrieved.
> > > > >
> > > > > Note that retrievals involve fewer distinct subjects and objects. It is because the LLM generation is often diverse and creative, proposing many combos of (subject, predicate, object) that involve ground-truth entities but haven't actually happened in real history. Such generated causes will eventually be grounded to actual past events that exhibit less diversity and creativity, thus ending up with fewer distinct subjects and objects in the retrieval.
> > > > >
> > > > > Therefore, as we can see, our overall framework is able to generate and retrieve the cause events that involve many attributes (predicates, subjects, objects) that are not mentioned in the few-shot examples. It means that our framework is really not bound by any list possibly mentioned in the prompts and has a great potential to generalize to a wide range of applications.
> > > > >
> > > > > > If the answer is yes, then why can an LLM guess reasonable predicate names without seeing them beforehand?
> > > > >
> > > > > Because LLMs have read nearly the entire internet and exhibit strong memorization and reasoning abilities. Since GPT-3 in 2020, people have found that LLMs can do many many challenging tasks very well after seeing only a few demonstrations (i.e., few-shot in-context learning). The recent GPT-4 is able to do many challenging tasks incredibly well with even no demonstrations (i.e., zero-shot in-context learning). This is our initial motivation, as stated in the paper, to pair an event sequence model with a large language model to improve its prediction accuracy. If an LLM can learn to perform well on GRE and LSAT and AP tests and coding competitions (as shown in GPT-4 Technical Report) after reading only a few examples, it seems reasonable to explore how well it will do on reasoning about possible cause events after reading only a few examples, isn't it?
> > > > >
> > > > > Precisely, in our problem setting, we think that LLM should have understood the basic meanings of the possible predicates (i.e., linguistic verbs) and have known many of the political entities during its pretraining and fine-tuning. Once a proposal is given, following the prompt of instruction and demonstrations, an LLM would be able to mimic the demonstrations and suggest possible causes, by utilizing its internal knowledge about the predicates and entities. This is just like how an LLM (e.g., GPT-3 and ChatGPT) performs any linguistic or reasoning tasks by in-context learning.
> > > > >
> > > > > Furthermore, LLMs have been shown to exhibit many impressive capabilities in recent literature in AI, ML, and NLP. The "emergence" phenomenon is inspiring and there has been work trying to demystify it. Progress in this direction will help us understand the empirical findings in our paper, but it is orthogonal and out of the scope of this paper.
> > > > >
> > > > > Have we resolved all your concerns (which are inspiring) now?

---

> > > > > ### Author Response · Authors · 2023-08-18
> > > > > **Have we resolved all your concerns?**
> > > > >
> > > > > Thank you again for your continued feedback.
> > > > >
> > > > > We deeply appreciate your participation in the discussion, and warmly welcome any further questions.
> > > > >
> > > > > As our author-reviewer discussion is coming to its end, we are looking forward to knowing whether our responses have resolved all your concerns so far, and whether you would like to adjust your evaluation. We hope that, with so many concerns addressed, you do not perceive our work as "a paper with technical flaws, weak evaluation" any more.

---

> > > > > > ### Comment · Reviewer_Hwvw · 2023-08-19
> > > > > >
> > > > > > Thanks. The detailed responses addressed all my main concerns. I increase my score to weak accept and expect the authors would make a careful revision especially on presenting formulae.

---

> > > > > > > ### Author Response · Authors · 2023-08-19
> > > > > > > **Thank you!**
> > > > > > >
> > > > > > > Thank you very much! We are excited to hear that our discussion has led to your positive evaluation of the work.
> > > > > > >
> > > > > > > We will carefully improve our presentation (esp. for math) for the final version.
> > > > > > >
> > > > > > > In the next day or two, we will post our (ongoing) plans for improving the paper.

---

### Official Review · Reviewer_HaQW · 2023-07-06

**Soundness:** 3 good
**Presentation:** 3 good
**Contribution:** 3 good
**Rating:** 5
**Confidence:** 4

**Summary:**

The paper investigates the use of large language models (LLMs) to improve the accuracy of event sequence models. Specifically, the authors propose an abductive reasoning framework based on an LLM. First, the event model produces some proposals of predictions. Then, the LLM suggests some causes for each proposal based on annotated demonstrations. At this stage, a search module retrieves some events that match the causes and finally a scoring function gives the likelihood that the retrieved events could actually cause the proposal.

**Strengths:**

* The proposed method is sound and applicable in a wide range of practical settings
* The experimental evaluation is solid, with several analyses and ablations that help the reader understand the contribution of different design choices
* The paper is well written and easy to follow

**Weaknesses:**

* The results are not surprising, as the baselines do not rely on the same setting
* The paper should mention the limitation that it requires textual data for the LLM

**Questions:**

None

**Limitations:**

Limitations are discussed in the Appendix. The paper should mention the limitation that it requires textual data for the LLM.

---

> ### Author Rebuttal · Authors · 2023-08-10
>
> Title: Response to Reviewer HaQW
>
> Thank you for your constructive feedback. We have added new results for you; please see [New Baselines] and [New Dataset]. Now we'll answer your remaining questions.
>
> > The results are not surprising, as the baselines do not rely on the same setting
>
> Do you mean that the baselines are only event sequence models, without any cause-retrieving and reranking mechanism? If yes, you may like our new results presented in [New Baselines]. Some new baseline methods we implemented follow the general propose-retrieve-rerank pipeline of our proposed framework, and they differ in how to retrieve causes. Can they be considered to "rely on the same setting"? As you can see, our full framework outperforms all the new baseline methods.
>
> In addition, we'd like to point out that our original baseline event sequence models are strong and difficult to outperform. Moreover, from an NLP perspective, whether LLM could help predict the future is an open question. Therefore, we honestly didn't know the improvements for sure before we actually carried out the experiments.
>
> If you still think that they are "not surprising" after reading this rebuttal, we would really appreciate that you elaborate it so we could kickstart a more in-depth discussion.
>
> Also, if we misunderstand what you mean by "same setting", please advise so that we could give a more informed response to this concern and further improve paper.
>
> > should mention the limitation that it requires textual data for the LLM.
>
> Sure. We will include it in the final version.

---

> ### Author Response · Authors · 2023-08-18
>
> Thank you again for your review.
>
> Has our response addressed your concerns?

---

### Official Review · Reviewer_E9bj · 2023-07-11

**Soundness:** 3 good
**Presentation:** 3 good
**Contribution:** 3 good
**Rating:** 7
**Confidence:** 3

**Summary:**

Summary:
This paper investigates the effectiveness of large language models in reasoning and predicting event sequences. The authors propose a general framework that combines event sequence models with large language models for the task of event prediction. In this framework, an event sequence model proposes predictions on future events given the past, while a large language model assists by generating possible causes for each proposal. Experiments on two challenging real-world datasets (Amazon Review and GDELT) have shown that this framework significantly outperforms state-of-the-art event sequence models.

Contributions:
  1. Proposes a domain-agnostic framework that combines event sequence models with large language models for the task of event prediction
  2. Demonstrates the framework's ability to significantly outperform state-of-the-art event sequence models on two challenging real-world datasets (Amazon Review and GDELT).
  3. Conducts extensive ablation studies to test the framework's robustness to prompt design and hyperparameters.

**Strengths:**

  1. The idea of using abductive reasoning to improve event prediction is really a natural fit, interesting, and novel.  The authors have made such a nice connection between two NLP problems. The framework is also pretty general.
  2. Performance: The experimental results show that the proposed framework outperforms state-of-the-art event sequence models on two datasets, and the amount of improvement seems large from the chart.
  3. The paper is well written and easy to understand.



**Weaknesses:**

  * The literature review is a bit limited.  For example, the paper claims "We are the first—to the best of our knowledge—to integrate large language models into temporal modeling."  But this is not true, as there are already works that apply LLMs to temporal reasoning.  For example, please see paper "Generic Temporal Reasoning with Differential Analysis and Explanation (https://arxiv.org/pdf/2212.10467.pdf)" and other papers from Dan Roth's group.

**Questions:**

  1. The result presentation is a bit hard to read, can tables be presented instead of charts?  It's really hard to read especially at the regions where methods are close to each other.
  2. In figure 4, the shapes are really too close to each other that it becomes difficult for the readers to read the figures.  Either make it bigger or differentiate them in other ways.
  3. Its really surprising that gpt3.5 is so much better than davinci-003, given the general belief of reasoning abilities of these models are gpt4 > text-davinci-003 > gpt3.5.  Do the authors have an interpretation?  Was the prompt tuning done on the gpt3.5 and later used for davinci?
  4. Can the authors do some qualitative analysis on the generated event causes?  What happens if the generated event causes for all event predictions are not included in past events?

**Limitations:**

The authors did not discuss limitations of this work.  I mainly see two limitations:
  1. The use of closed-source model limits the applicability of this approach.
  2. It requires manually writing prompts for specific tasks.

---

> ### Author Rebuttal · Authors · 2023-08-10
>
> Thank you for your constructive feedback and being supportive! We have added new results to resolve your concerns; please see [New Baselines] and [New Dataset]. Now we'll answer your remaining questions.
>
> > The literature review is a bit limited… There are already works that apply LLMs to temporal reasoning…
>
> Thanks for reference! We apologize for our misleading claim. What we meant is: we are the first to apply LLMs to the area of event sequence modeling (specifically, the community discussed in [Contribution, Significance, and Impact], which I meant by "temporal modeling"); or more precisely, we propose the first framework that leverages LLMs to enhance event sequence models.
>
> We are well aware of the line of "temporal reasoning" work in NLP. We apologize for not discussing it in the submitted version. In final version, we will add a paragraph discussing the similarities and differences between our work and the "temporal reasoning" NLP work. This line of NLP work includes many papers (e.g., Feng et al. 2022 as you suggest) from NLP groups led by Dan Roth, Heng Ji, Kathleen McKeown, Muhao Chen, Qiang Ning, etc.
>
> > can tables be presented instead of charts? ... In figure 4, the shapes are really too close…
>
> Definitely! We'll keep figures in the main paper, but add the actual numbers in appendices. We will also make the curves and markers more distinguishable, by using larger sizes and different shapes.
>
> > really surprising that gpt3.5 is so much better than davinci-003… Do the authors have an interpretation? Was the prompt tuning done on the gpt3.5 and later used for davinci?
>
> We found it surprising, too. Based on our new results ([New Baselines]), it seems that RLHF is important for this problem setting: when using LLAMA2-chat-13B, our framework works competitive to the GPT-3.5 version, and significantly better than the GPT-3 version. It is unclear how RLHF has helped this particular problem, which seems to be a good future research topic.
>
> Moreover, we didn't tune prompts: we designed the prompt templates before running any experiments, and haven't changed it since then.
> Over the past week, we tried a few different prompt templates for each dataset, and found that it doesn't change the ranking of the methods (i.e., GPT-3.5 > LLAMA2-chat-13B > GPT-3). We'll add this new analysis in the final version.
>
> > What happens if the generated event causes for all event predictions are not included in past events?
>
> For each generated cause, the most relevant (measured by sBERT embedding similarity) will be retrieved; see Line-133. It is the duty of the ranking module to analyze each (proposal, retrieved evidence) combination and assign low scores to implausible combinations.
>
> In [New Baselines], we show results of "only retrieve events if its retrieval score is higher than threshold", but its performance is not very different from "always retrieve 10 highest-scored events, no matter how low they are".
>
> > ​​did not discuss limitations… the use of closed-source model limits the applicability… requires manually writing prompts
>
> Limitations section is in appendices. And we will add your points in the final version.
>
> In response to your point of "close-source LLMs", we add new results using open-source LLAMA2-chat-13B; please see [New Baselines].
>
> > Can the authors do some qualitative analysis
>
> We'll add them in the final version. See [Qualitative Results].

---

> > ### Comment · Reviewer_E9bj · 2023-08-15
> > **Thank you for the response**
> >
> > The author response has addressed my confusion, and I would like to see this paper accepted.  Event-based reasoning has remained as a great challenge in NLP, and this paper provides a really simple and effective technique to improve the performance.

---

> > > ### Author Response · Authors · 2023-08-17
> > > **Thank you for your support!**
> > >
> > > Thank you very much for your support! And we really appreciate your acknowledgement that event-based reasoning is a great challenge in NLP.
> > >
> > > We have been improving the paper presentation based on all the reviews, and we will surely deliver a strong and high-quality camera-ready to NeurIPS.
> > >
> > > Later in the Discussion phase, we will post our full list of presentation improvements. (We will hold it until then in case that any reviewer requests anything new.)

---

### Author Rebuttal · Authors · 2023-08-10


We thank reviewers for constructive feedback! In this to-all message, we clarify our technical contributions and present new results. We will address other concerns in responses to individual reviews.

Due to 6000-char limit, we have to keep this response concise. If anyone asks for more details, we'd be happy to elaborate them in Discussion phase.

[Contribution, Significance, and Impact]

Event sequence modeling (i.e., learning to predict future events given the past) is an important area in machine learning. In recent years, over a thousand new papers on this topic have been posted on arXiv. Our paper makes a significant contribution to this area: we propose a simple modeling and inference framework, which leverages recent advances in NLP---LLMs and in-context learning---to improve prediction accuracies of traditional event sequence models. Our framework is compatible with any event sequence models. Our experiments show that it significantly and consistently improves over a range of strong and influential event sequence models. This finding is certainly of great interest to both event sequence modeling and NLP (particularly, LLM) research communities, which is a broad audience. In addition, we will release code to assist researchers in reproducing our results and conducting their future research.

R.mHgh thinks that "a more worth-studying problem is how to use LLMs to more accurately predict future events in the first place". We agree it is "worth-studying", but it is orthogonal to our contribution. Moreover, "which is more worth-studying" is subjective: in history of science, many problems were regarded as "not important", but some of them turned out to be extremely important later in time.

[Scientific Rigor]

R.Hwvw is concerned with scientific rigor and soundness of this paper, and asks for more "theoretical analysis and empirical study". Please see [New Baselines] and [New Dataset] for new results and analysis.

As for "theoretical analysis", we'd like to emphasize: some papers are theoretically rigorous; some are empirically rigorous. This paper is empirically rigorous: we carefully thought through the framework design, evaluated against strong and proper baselines, and conducted extensive empirical studies. Our method has no theoretical guarantees, but this limitation is shared by many great empirical papers and should not be a reason for rejection.

Surely, there could be settings where our framework is no better than baselines: e.g., LLMs provide irrelevant, incorrect, or misleading information. We'll discuss such cases in the Limitations section.

[About Experiment Design]

R.mHgh is concerned with our experiment design. For a fair comparison with previous work, this paper follows standard experiment setup:
- evaluate on predicting time and type of future events, given the history;
- only predict an attribute of type (e.g., predicate) when there are too many types.

GDELT has over 20 millions of event types (in format of subject-predicate-object). First, full type prediction is extremely hard and all methods will end up with indistinguishably low accuracies. Second, in real applications, users usually care about "what will happen between entity A and entity B" (predicate prediction) and "what will A do to B" (predicate-object joint prediction), instead of "which of 20M possible events is most probable" (full type prediction). Therefore, previous work using GDELT also evaluates on attribute prediction.

[New Baselines]

We ran experiments with 7 new baselines. They include: 3 methods that follow the pipeline of our framework but differ in how to retrieve evidence; a new version of ours that uses LLAMA2-chat-13B; a new event sequence model that uses text information; a method that directly uses LLMs for event prediction; a new version of ours that retrieves a past event only if its retrieval score is higher than a threshold.
Results are in Fig-16 and 17 of submitted pdf. Takeaways are:
- our GPT-3.5-based framework outperforms all the new baselines.
- LLAMA2-chat-13B version of our framework is better than GPT-3 (175B) version, indicating that RLHF may be important for our problem setting. This finding is interesting and worth studying in the future.
- directly using LLMs for future prediction performs worse than base ANHP model.
- there is no much difference between "retrieve 10 highest-scored events" and "retrieve if it passes threshold".

[New Dataset]

We ran experiments on ICEWS, a dataset similar to GDELT but less dense in time (so that time prediction is more meaningful).
Results are in Fig-18 of submitted pdf: like on GDELT and Amazon Review, our LLM-enhanced framework significantly outperforms baselines on ICEWS.

[Qualitative Results]

LLM-generated causes look reasonable and meaningful. We'll show them in the final version.

[Limitations]

Our Limitations section is in appendices. In final version, we'll discuss the additional limitations pointed out by reviewers. We already wrote a new version, but couldn't share it here due to char limit. If needed, we can show it in Discussion phase.

However, we kindly request reviewers to be mindful of our existing contributions and significance. After all, what hasn't been done should not undermine the value and significance of what has already been done.

[Presentation Improvements]

We will include new results in final version. In addition, we have planned a list of presentation improvements, including ways to fix issues pointed out by reviewers (e.g., notation definition, formula references). With extra page allowed by NeurIPS camera-ready, the fixes are easy to execute. Considering that we have managed to deliver many new results within a week, we hope that reviewers feel assured of our promise and commitment to complete the proposed improvements. If needed, we'd like to show our full list of presentation improvements in Discussion phase, for reviewers to check and approve.

---

### Author Response · Authors · 2023-08-20
**Details of Presentation Improvements**

We thank reviewers again for their active participation in the discussion. We are thrilled that the discussion will lead to a final version of improved quality.

Here, as promised in previous messages, we present a list of presentation improvements that we will execute for the camera-ready. We ask AC and reviewers to stay reassured and not worry about the length of the list: they are all easy and straightforward, based on the comprehensive discussion between reviewers and us. There will be a lot of new results and analysis, but they will be easy to accommodate with the extra page allowed by NeurIPS camera-ready. Actually, we have already rewritten several parts of our paper, clarifying our contributions as well as incorporating new results and discussion. We are whole-heartedly committed to delivering an as good camera-ready as we could.

Below is our list:
- clarifying our contributions. In Abstract and Introduction, we will add content explaining that event sequence modeling has been a long-standing and challenging research topic in ML, and that we aim to advance it by introducing LLMs and in-context learning.
- clarifying our technical details. First, we will clarify our notation system and properly label math formulas (and explain their intuition) in Sec 2 to facilitate reference and discussion in later sections. Second, at the beginning of Sec 3, we will give a technical overview of our entire framework before Sec 3.1: the current version (up to line 94) only explains how an event sequence model proposes predictions, and why they may not be correct; we will add sentences highlighting how an LLM may help rerank the predictions through its reasoning capabilities (such insights are scattered in Sec 3.1, 3.2, and 3.3 in the current version).
- discussing classical logical reasoning. We will add a paragraph discussing the deep connections between our method and classical logical reasoning methods (e.g., backward chaining). A current draft can be seen in our response [More About Logical Reasoning] to R.pkDi.
- discussing "temporal reasoning" NLP work. We will add a paragraph in Related Work, discussing the NLP work in temporal reasoning as well as the technical connections between our work and that line of research.
- explaining our prompt design. E.g., we will emphasize (with experiment results) why our method can apply to a wide range of applications with a large number of event types.
- justifying our experiment design. In Sec 5.1, we will introduce our experiment and evaluation design (e.g., attribute prediction), explaining why it is a standard setup and provides a fair comparison.
- clarifying experiment details. We will add more experiment details (e.g., history includes all the previous events), and clarify interpretation of results (e.g., as discussed with R.mHgh).
- adding new experiment results and analysis. We will add all the new results and analysis presented in the rebuttal and discussion. For some of them (e.g., new ICEWS dataset, new MRR eval metric), we will complete a fuller set of experiment results (e.g., evaluating all baseline methods on ICEWS and making it as full as GDELT; evaluating new baseline methods on existing Amazon data; adding MRR evaluation for all experiments).
- improving accessibility of results. We will adjust the figures so they are more visually accessible. In addition, we will provide the actual numbers (of elements in figures) in appendices for future reference and comparison.
- adding qualitative results. We will show LLM-generated causes as well as retrieved evidence.
- discussing limitations and future work. We will discuss all the technical limitations pointed out by reviewers, as well as our thoughts on how future work may break these limitations. We will also discuss possible extensions of our work (e.g., multiple-step reasoning, joint optimization of modules, generalization to other settings with non-temporal facts/events).

---

### Decision · Program_Chairs · 2023-09-21

**Decision:**

Accept (poster)

**Comment:**

This paper demonstrate that language models can be employed to assist event sequence models for even prediction. The framework proposes using an event sequence model to propose potential events, then use a language model to retrieve past events that could help in predicting which event is more likely in the future and outputting likely events and finally using a scoring mechanism to find the most likely event. The paper is interesting, well-written and easy to follow. The contributions are significant as they open up the opportunity for more LLM-related research in this area. All reviewers voted for acceptance. Therefore, I recommend acceptance.